# EduMirror: Modeling Educational Social Dynamics with Value-driven Multi-agent Simulation

## Abstract

The scientific study of educational social dynamics, such as bullying and peer pressure, is crucial for student well-being yet hindered by profound ethical and methodological barriers inherent in traditional research. While multi-agent simulations powered by Large Language Models (LLMs) provide an ethically viable alternative, they often fail to bridge the gap from believable narratives to rigorous experiments, plagued by two fundamental hurdles: a lack of psychologically plausible motivations (the Fidelity Challenge) and the absence of systematic methods for quantifying complex interactions (the Measurement Challenge). To overcome these obstacles, we introduce **EduMirror**, a multi-agent simulator for the scientific study of educational social dynamics. EduMirror integrates four key components: (1) A Systematic Scenario Design Workflow grounds simulations in established social science theory, ensuring construct validity. (2) To address the Fidelity Challenge, a unified Value-Driven Agent Architecture models agent motivation based on both individual psychological needs and Social Value Orientation (SVO). (3) To solve the Measurement Challenge, a Dual-Track Measurement Protocol employs specialized LLMs as a post-hoc Rater for observable behaviors and an in-situ Surveyor for internal states, transforming qualitative interactions into quantitative data. (4) Together, these components enable researchers to conduct controlled Intervention Experiments, branching simulations to systematically assess the causal impact of different strategies. We validate our platform through case studies on school bullying and group cooperation, demonstrating that the framework can generate social phenomena aligned with established theories and measurable through empirical criteria, suggesting a feasible pathway toward structured in silico educational research.

## 1 Introduction

The educational environment is a crucible for adolescent development, where social and emotional dynamics such as school bullying and peer pressure act as critical determinants of student well-being and lifelong outcomes Hymel & Swearer (2015). These complex phenomena are not peripheral to academic learning; they are central to it. Mounting evidence establishes that experiences like bullying are not harmless rites of passage but severe public health issues, inflicting deep and often irreversible psychological and physiological scars Wolke & Lereya (2015); Arseneault (2018). Landmark studies have found that the long-term mental health consequences of peer bullying can be even more severe than those of adult maltreatment, positioning it as a profound form of childhood adversity Takizawa et al. (2014). This reality imparts a profound moral imperative to understand and mitigate these harmful dynamics, as the cost of ineffective interventions is unacceptably high.

This pressing social imperative confronts researchers, educators, and policymakers with a formidable ethical dilemma. The scientific gold standard for establishing causality, the Randomized Controlled Trial (RCT), is ethically impermissible for studying the unmitigated effects of harmful phenomena. It is not feasible, under the guiding principles of the Belmont Report, to assign students to a "no-intervention" control group to observe the pure impact of bullying for the Protection of Human Subjects of Biomedical & Research (1979). Compounding this challenge, even well-intentioned interventions carry the risk of iatrogenic harm, where programs inadvertently worsen

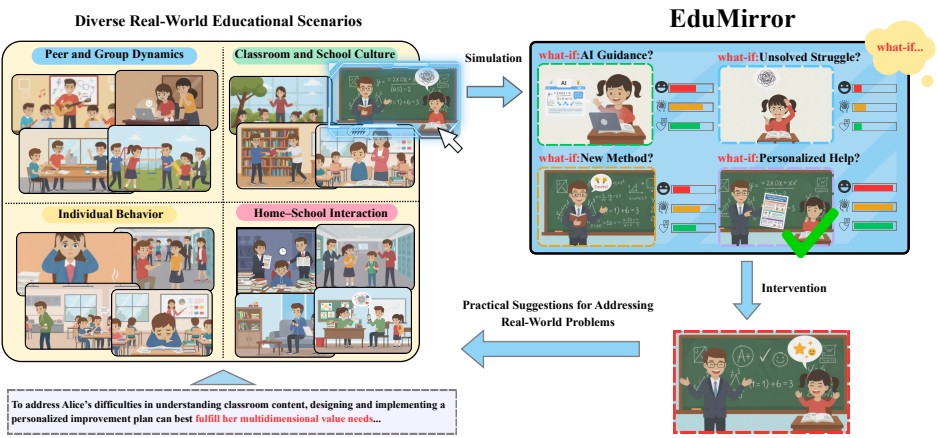

Figure 1: An illustration of the core concept behind *EduMirror*. Like a mirror, EduMirror simulates a wide range of authentic educational scenarios, enabling reflection on real-world practices and projecting the potential outcomes of different interventions. By modeling the individual and social values of agents across multiple dimensions, it aims to maximize their fulfillment and generate practical, actionable insights for real-world educational challenges.

student outcomes through mechanisms like deviancy training or stigmatization Foulkes & Stringaris (2023). Traditional observational methods, such as self-report surveys, offer a safer alternative but are notoriously compromised by recall and social desirability biases, especially on sensitive topics Latkin et al. (2017); Perreault (2017); Latkin et al. (2016). These methods provide static, correlational snapshots and fail to capture the generative mechanisms of social interaction Shiffman et al. (2008).

To escape this ethical and methodological impasse, we turn to a third paradigm of scientific inquiry: *in silico* experimentation. This approach, rooted in the philosophy of generative social science, posits that to truly explain a social phenomenon is to "grow" it from the bottom up through the interactions of heterogeneous agents Epstein (2006). We propose the concept of a 'digital mirror', a computational laboratory designed not merely to reproduce surface narratives, but to test the causal and motivational mechanisms underlying educational social dynamics. This paradigm is well-established in other high-stakes domains. Just as climatologists use computational models to test policies in a digital Earth Schneider (2009) and engineers use "digital twins" to manage critical urban systems Grieves & Vickers (2017); Marçal-Russo et al. (2025), educators require a similar tool to safely, ethically, and repeatedly explore "what-if" scenarios that are forbidden in reality.

However, constructing a digital mirror of sufficient scientific integrity presents a grand challenge. The leap from creating believable narratives to conducting rigorous, replicable experiments faces two fundamental hurdles that have long plagued the social sciences. The first is the **Measurement Challenge**: many of the most critical effects of social dynamics occur in students' internal psychological states (e.g., self-esteem, sense of safety), which are inherently difficult to observe and quantify reliably Perreault (2017); Latkin et al. (2016). The second is the **Fidelity Challenge**: to accurately model the emergence of complex social behaviors, simulated agents must be driven by deep, psychologically plausible motivations, not by the brittle, hand-crafted rules characteristic of traditional Agent-Based Modeling (ABM) Bordini et al. (2016). This requires a framework that can bridge the longstanding trade-off between the internal validity of controlled lab experiments and the ecological validity of real-world observation Bronfenbrenner (1977); Schmuckler (2001).

To bridge this crucial gap, we introduce EduMirror, a multi-agent simulation platform designed for the scientific study of Educational Social Dynamics. Our core technical approach establishes an end-to-end computational experimentation framework that spans from theory-driven scenario design and high-fidelity simulation execution to user-led causal intervention and multi-dimensional result analysis. Through this framework, we make four key contributions:

**1) A Systematic Scenario Design Workflow.** We establish a rigorous five-step protocol that serves as the cornerstone of all our simulations. This workflow systematically transforms an abstract educa-

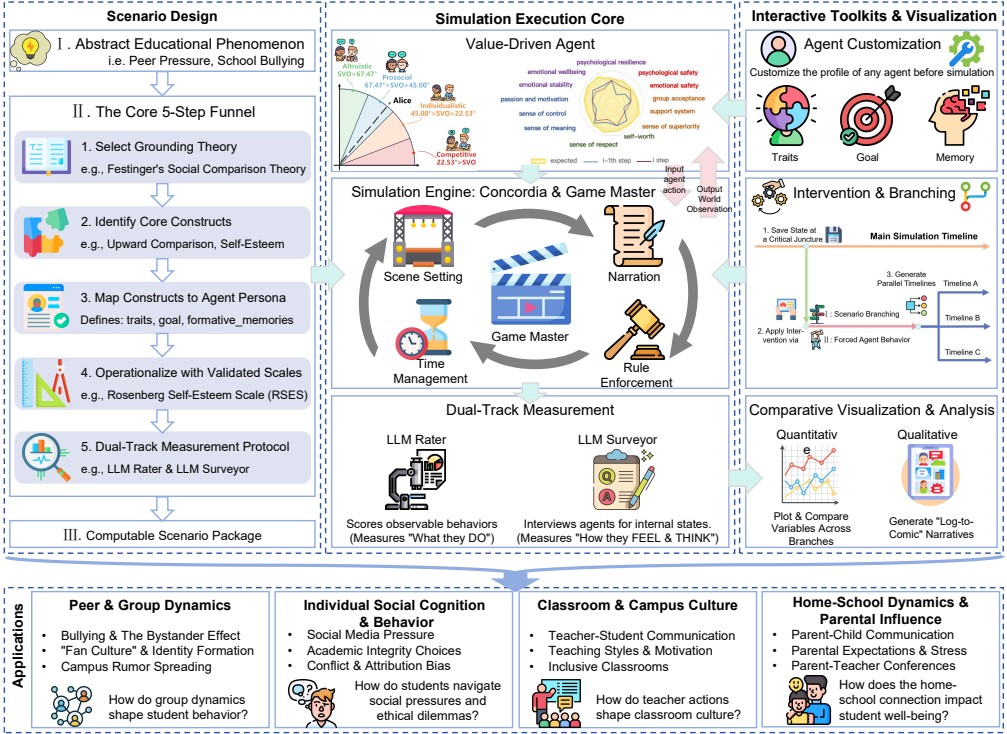

Figure 2: The architecture of EduMirror, our multi-agent simulation platform. The research workflow proceeds through three main stages. (Left) The Scenario Design module employs a five-step, theory-grounded process to convert an educational phenomenon into a computable scenario. (Center) The Simulation Execution Core executes the scenario, integrating value-driven agents, an environment orchestrated by a Game Master, and a dual-track measurement protocol (LLM Rater and Surveyor). (Right) The Interactive Toolkits & Visualization module enables user-driven experimentation through agent customization, intervention and branching for comparative analysis, and tools for both quantitative and qualitative visualization. (Bottom) The Applications panel illustrates the platform's capacity to investigate various educational challenges across four key domains.

tional phenomenon into a design that is both scientifically rigorous and computationally executable, ensuring the validity and reproducibility of the experiments.

**2) A Dual-Track Measurement Protocol.** To transform the rich, qualitative interactions within the simulation into reliable quantitative data, we introduce a measurement protocol that employs two specialized LLM assessors. One, an LLM Rater, performs post-hoc analysis of observable actions, while the other, an LLM Surveyor, conducts in-situ probing of internal states. This approach captures both behavioral and psychological dynamics that are traditionally difficult to measure.

**3) A Unified Value-Driven Agent Architecture.** To achieve high behavioral realism, we design a unified agent architecture with an intrinsic motivational structure. This architecture can be configured with one of two parallel value systems: an Individual Value system, grounded in psychological need theories to model well-being and stress Ryan & Deci (2000); Maslow (1943a), or a Social Value system, based on Social Value Orientation (SVO) theory to model decision-making in social dilemmas Murphy et al. (2011); Van Lange (1999). This ensures agent behavior is driven by deep, theoretically-informed psychological dynamics.

**4) An Interactive Environment for Causal Experiments.** We engineer EduMirror as an active computational laboratory where users can not only customize agents but also apply interventions during the simulation. This capability transforms the simulation from passive observation into a platform for controlled causal experiments, allowing researchers to systematically test the effectiveness of different strategies.

## 2 EDUMIRROR

To systematically investigate complex educational phenomena through computational experiments, we have developed EduMirror, a modular and interactive multi-agent simulation platform. The architecture of EduMirror, illustrated in Figure 2, is designed to support a structured research process encompassing scenario design, simulation execution, and interactive analysis. This section details the primary components of the platform. To facilitate understanding, we use a single, comprehensive example to illustrate the entire simulation process, as detailed in Pre-designed Scenarios in EduMirror.

### 2.1 SYSTEMATIC SCENARIO DESIGN WORKFLOW

The foundation of EduMirror is a systematic, five-step workflow that translates abstract educational phenomena into computable scenarios, as detailed in Figure 2. The process begins by (1) **selecting a grounding theory** (e.g., Social Comparison Theory) to anchor the scenario scientifically. Next, we (2) **identify core constructs** by deconstructing the theory into fundamental concepts, which then (3) **guide the agent persona configuration**, where we initialize agent `traits`, `goals`, and `memories` to reflect the chosen theoretical model. To ensure empirical rigor, we (4) **operationalize these constructs with validated scales** (e.g., RSES), and finally (5) **establish a dual-track measurement protocol** using LLM Raters and Surveyors to quantify agent behaviors and internal states. This structured approach ensures that experimental outputs connect back to specific theoretical constructs. A detailed walkthrough is available in Pre-designed Scenarios in EduMirror.

### 2.2 AGENT ARCHITECTURE

Agents in EduMirror are designed to capture multiple facets of human motivation. The platform supports agent customization prior to simulation, allowing users to modify an agent's personality `traits` (e.g., using MBTI or Big Five models), core `goal`, and formative `memories` (see the Agent Customization panel in Figure 2). This functionality enables the systematic exploration of how individual characteristics influence outcomes. The behavior of each agent is driven by one of two selectable models, depicted in the Value-Driven Agent portion of Figure 2:

**Individual Value Model (Psychological Needs)** This model is used for scenarios examining individual well-being and stress responses.It builds on the D2A frameworkWang et al. (2024b), which focuses on driving an agent to generate human-like activities based on human needs in the absence of explicit task instructions. It consists of two core modules: the Value System and the Desire-driven Planner. The Value System manages desire components, each representing the level of satisfaction for a desire dimension. During the simulation, these desire components are initialized with initial and expected values, and five key steps are performed: Qualitative Value Description, Activity Proposal, Activity Evaluation, Activity Selection, and Need Value Update. The goal of the agent is to execute appropriate activities that align its desire components with the expected values.

The simulation consists of $T$ time steps. At each time step $t$, the agent generates a new activity $a_t$ based on the given context, including past activities $a_0 : t - 1$, observations $o_0 : t - 1$, customized information $I$, agent profile $P$, environment description $e$, and agent parameters $\theta$ (e.g., LLM):

$$a_t \sim \text{Agent}(|a_0 : t - 1, o_0 : t - 1, I, p, e; \theta)$$

After $T$ steps, the activities generated by the agent are collected and rewritten into a coherent sequence.

What differentiates our model is that it focuses on human intrinsic psychological states, referring to Maslow's Hierarchy of Needs Maslow (1943b) and the PERMA model from Positive Psychology Seligman (2011). We expand the Value System into a Psychological Need System, which simulates deep human psychological drivers through five major categories of psychological needs (Safety Needs, Mental Health Needs, Self-Esteem Needs, Social Belonging Needs, Meaning and Growth Needs) and a total of 13 subdimensions. Each need dimension is scored using a Likert scale, with values ranging from 0 to 10. As shown in the table 6 of our ablation studies, each category of psychological needs plays an important role in the agent's ability to generate coherent, natural, and

plausible behaviors.We also considers the impact of personality traits on the expected values of psychological needs. During the initialization phase, corresponding psychological need values are automatically mapped based on personality traits (see Table 5 in Appendix D).

Additionally, the Desire-driven Planner in our model has been extended to a Need-driven Planner, adding explicit procedures and prompts during the candidate behavior generation process to drive the agent to simulate the most likely natural reactions of a human under its current psychological state and environmental context. These behaviors encompass a broader range of external expressions, such as emotional reactions, physical movements, and verbal expressions. The system then evaluates how well each candidate behavior matches the current psychological state, considering potential changes in need values after executing the current response. The agent does not aim to achieve a specific need value; instead, the behavior selection module chooses the behavior that provides a reasonable response to a broader range of psychological need dimensions, which refers to the option that better aligns with the agent's psychological needs, and uses it as the final behavior in the current context.Further details and related prompts are provided in Appendix D and F.

**Social Value Model (SVO)**   EduMirror models cooperation and competition through a principled Social Value Orientation (SVO) formulation. Each agent is initialized with a target SVO type (Altruistic, Prosocial, Individualistic, Competitive), which specifies its theoretical preference interval. During interaction, the agent's moment-to-moment SVO is not fixed but continuously recalculated based on changes in its internal motivational state and its inference about others.

At each step, the agent evaluates how well its psychological needs are being met by comparing the current magnitude of each desire $v_t(d)$ with its expected level $v^*(d)$:

$$\Delta_t(d) = v^*(d) - v_t(d).$$

These deviations are aggregated into satisfaction scores for the agent itself and for other agents, reflecting whether recent interactions improve or reduce motivational alignment. Following the standard definition of SVO, the agent determines its current social preference orientation as:

$$\theta_t = \arctan\left( \frac{S_{\text{other}}(t) + \varepsilon}{S_{\text{self}}(t) + \varepsilon} \right),$$

where $S_{\text{self}}(t)$ and $S_{\text{other}}(t)$ denote self and other satisfaction, each clipped to reflect bounded human perception. This formulation captures how relative improvements in others' outcomes versus one's own yield shifts in cooperative or competitive tendencies. Finally, given predicted utilities for self and others under each candidate action,

$$U(a) = \cos(\theta_t)\, U_{\text{self}}(a) + \sin(\theta_t)\, U_{\text{other}}(a),$$

the agent selects $a_t = \arg\max_a U(a)$. This SVO-guided utility integration ensures that decision-making consistently reflects both the agent's evolving social orientation and its underlying personality-defined SVO interval.

### 2.3 SIMULATION ENVIRONMENT AND USER INTERVENTION

**Simulation Environment and the Game Master**   The environment is powered by the Concordia library and orchestrated by a central Game Master (GM), as shown in the Simulation Engine diagram in Figure 2. The GM has four responsibilities: setting the initial scene, narrating world events, enforcing rules, and managing time. This centralized control structure is designed to support the reproducibility of experiments.

**Intervention and Branching**   A key feature of EduMirror is the ability to conduct comparative experiments from a single simulation run. As outlined in the Intervention & Branching panel of Figure 2, the process begins when a user saves the complete state of a simulation at a critical juncture along the main timeline. From this saved state, the user can apply an intervention to generate multiple parallel branches for comparison. Interventions are applied in two primary forms to test causal impact. With Scenario Branching, a user alters the narrative path by introducing a new event or modifying the environment, effectively choosing a different direction for the story to unfold, much like following a new path at a signpost. For instance, a new timeline can be created where a teacher initiates a

supportive conversation, allowing researchers to study the impact of this contextual change. Alternatively, Behavior Control allows the user to act as a puppeteer, directly dictating a specific agent's action for a single step and overriding its autonomous decision-making. This powerful technique enables a precise examination of the direct consequences of a single behavior. Following the intervention, the platform generates parallel timelines, enabling direct, counterfactual comparisons to test the causal impact of different strategies and actions.

### 2.4 MEASUREMENT AND ANALYSIS

**Dual-Track Measurement Protocol**  To quantify agent states and behaviors, we employ a measurement protocol utilizing two LLM-based assessors, as shown in the Dual-Track Measurement section of Figure 2. The LLM Rater functions as a post-hoc analyzer, systematically scoring observable behaviors from interaction logs. Concurrently, the LLM Surveyor acts as an in-situ interviewer, posing psychometric questions during the simulation to probe their internal states.

**Comparative Visualization and Analysis**  Following the generation of parallel timelines, EduMirror provides tools for analysis, depicted in the Comparative Visualization & Analysis panel of Figure 2. For quantitative analysis, the platform generates plots comparing key variables across different experimental branches. For qualitative analysis, a "Log-to-Comic" feature visualizes simulation logs as a comic strip, offering an intuitive narrative representation of emergent dynamics.

### 2.5 APPLICATIONS AND SCENARIOS

The modular architecture of EduMirror supports a wide range of computational experiments in education, as summarized in the Applications panel of Figure 2. The platform's versatility stems from its diverse simulation environments and ability to model various complex social phenomena.

**Scenarios**  EduMirror provides eight pre-configured virtual environments that represent key locations in a student's life. These include the *classroom*, *dormitory*, *playground*, *cafeteria*, *home*, *teacher's office*, *gymnasium*, and *library*. This variety of settings enables the simulation of phenomena that span school and home contexts, to better investigate educational issues.

**Applications**  Within these scenarios, EduMirror is used to investigate 20 applications across four main themes (Peer & Group Dynamics, Individual Social Cognition, Classroom Culture, and Home-School Dynamics). These applications address key issues such as bullying and bystander effects, materialistic social comparison, teacher burnout, and the impact of different parenting styles.

## 3 EXPERIMENTS AND RESULTS

To validate the methodological contributions of EduMirror, we present two distinct case studies. The first case study leverages the **Individual Value Model** to simulate the complex psychological dynamics of school bullying; the second draws on the **Social Value Model**, grounded in Social Value Orientation (SVO), to demonstrate that the platform can generate emergent, theory-consistent patterns of cooperation and competition. Building on this, we also conducted intervention experiments in both case scenarios, showcasing the platform's ability to simulate and analyze the effects of different intervention strategies in educational contexts. The source code and scenarios are available at https://anonymous.4open.science/r/EduMirror.

### 3.1 CASE STUDY 1: SCHOOL BULLYING SIMULATION

In this case study, we present a series of experiments conducted within a controlled school bullying simulation environment to address two key research questions: (1) Can EduMirror reproduce school bullying scenarios that resemble real-world incidents in both dynamics and narrative coherence? (2) Do agents modeled under our individual value framework generate more emotionally dynamic and human-like behaviors compared to baseline approaches? Building on these investigations, we further explore the application of our platform in conducting intervention experiments to evaluate the psychological effects of different teacher response strategies.

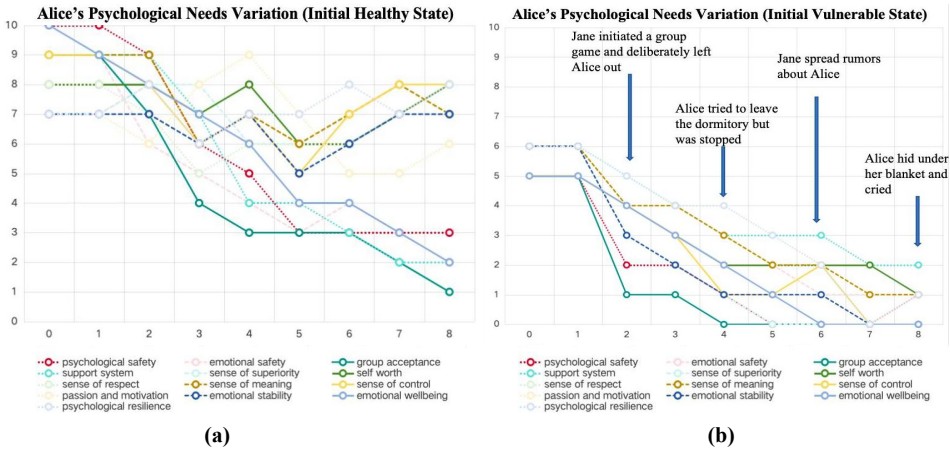

(a)                                            (b)

Figure 4: Comparison of the dynamics of psychological needs under different initial states in the dormitory bullying scenario. The vertical axis represents value scores (0–10), the horizontal axis denotes time steps (each corresponding to 20 minutes), and different curves indicate distinct psychological need dimensions.

**Evaluation of Simulation System**  We conducted a series of bullying simulations using EduMirror under different initialization conditions to reproduce bullying interactions and capture victim responses. The bully agents exhibited a wide range of behaviors, with frequencies varying across contexts. Specific details can be found in Appendix E.

To assess the realism of our simulation, we conducted a survey to see if people could distinguish our simulated bullying cases from real ones. We paired ten real cases, sourced from online news and interviews, with ten simulated cases of similar settings. All cases were rewritten in a unified style using GPT-4o. The survey, shared online and via social media, received 152 valid responses. Participants were asked to identify the real case or select "difficult to distinguish."

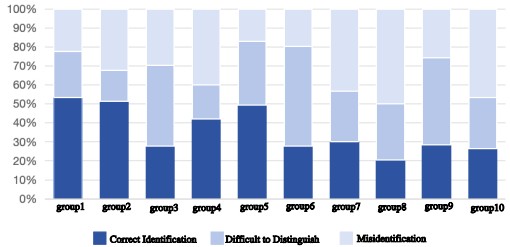

Figure 3: Results of the questionnaire survey. Overall accuracy in distinguishing real from simulated cases was low, with several simulated scenarios frequently misidentified as real, indicating the high realism of the generated bullying events.

As shown in Figure 3, participants had low accuracy in distinguishing real from simulated cases. Groups 1 (53.29%) and 2 (51.32%) slightly exceeded chance, while most groups scored below 30%, with Group 8 at the lowest (20.39%). Misclassification was common, particularly in Group 8 (50.00%) and Group 10 (46.71%), where simulated cases were often judged as real. Many also chose "difficult to distinguish" (e.g., Group 6: 52.63%). These results suggest that our system generates highly realistic and coherent bullying scenarios.

**Evaluation of Individual Value Model**

To assess the realism of our model in simulating victims' psychological dynamics in school bullying, we compared it with three baselines: ReAct Yao et al. (2023a), LLMob Wang et al. (2024a), and BabyAGI Nakajima (2023a). Fifteen bullying scenarios were created, with each model playing the victim role under identical conditions. GPT-4o was used as an external evaluator to assess activity sequences on three dimensions: naturalness, coherence, and plausibility (see Appendix E for details).

The win-rate heatmap (Figure 5) shows that our model outperformed all baselines, demonstrating a stronger ability to generate human-like behavior in bullying scenarios. GPT-4o's evaluations were closely aligned with human judges (see Appendix E). Our model was favored for its "comprehensive psychological response pathways" and "natural emotional expressions," while ReAct was deemed

"idealized," BabyAGI as a "static victim," and LLMob criticized for lacking emotional depth and contextual integration.

This enhanced ability to simulate diverse behaviors and emotional responses stems from the fact that the victim agent, Alice, modeled under our individual value framework, exhibited dynamic fluctuations in her values across different contexts. This variability mirrors real-world psychological state changes, leading to diverse behaviors and emotional responses. As shown in Figure 4, Alice's initial values strongly influenced her coping strategies and the progression of bullying scenarios. Higher initial values resulted in greater resilience and psychological stability, while lower values led to increased emotional volatility, accelerating the bullying.

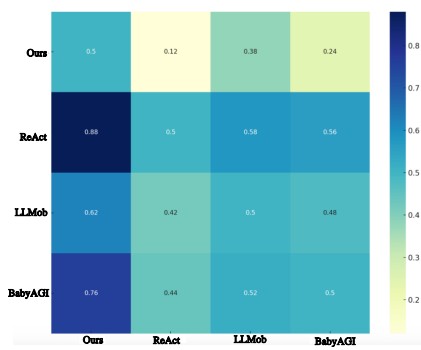

Figure 5: Win-rate heatmap of pairwise comparisons among models. Our model consistently outperformed baselines, indicating superior human-likeness in simulated bullying scenarios. Each cell indicates the win rate of the column model relative to the row model in pairwise comparisons.

**Intervention Experiments**

Teachers play a vital role in school bullying as their interventions influence both the course of incidents and the recovery of victims. Previous studies highlight three main strategies: (a) *authoritative punitive*, (b) *supportive individual*, and (c) *cooperative support*, with the cooperative approach being the most effective Seidel & Oertel (2017); Wachs et al. (2019). To assess the psychological impact of these strategies, we introduced a "teacher" agent under four conditions: three intervention types and a no-intervention control, and created 20 bullying scenarios with identical initial settings. Teacher agents with different goals generated distinct behaviors (see Table 10 in Appendix E). We then compared how each strategy influenced changes in the victim agent Alice's psychological values, as shown in Figure 6.

The results reveal a clear progression in intervention effectiveness, from *ignoring* to *authoritative-punitive*, *supportive-individual*, and finally, *supportive-cooperative*, which proved most effective. When ignored, victims showed a consistent decline in all psychological needs, especially safety and belonging, reflecting a lack of emotional support. The authoritative-punitive approach showed modest improvements in safety, belonging, and mental health but had limited or negative effects on self-esteem and meaning. The supportive-individual strategy led to moderate gains, particularly in safety and mental health, though its effects on social connection and agency were inconsistent. The supportive-cooperative approach resulted in the most significant improvement across all psychological need dimensions, highlighting the importance of collective actions from peers, teachers, and families for both immediate emotional support and long-term well-being.

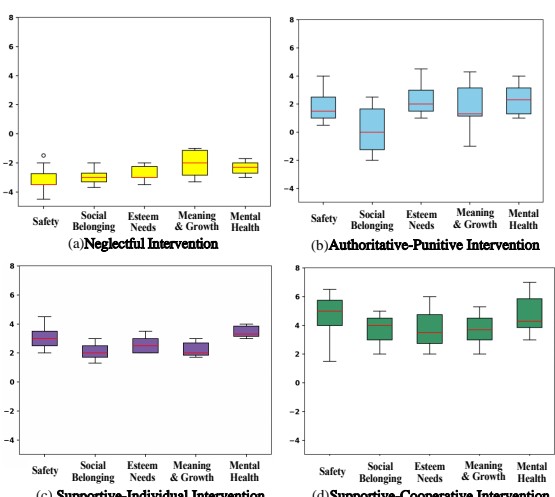

Figure 6: Changes in victims' values under different interventions. The change in each major psychological dimension was calculated as the arithmetic mean of the value changes across its subdimensions.

## 3.2 CASE STUDY 2: EMERGENT SOCIAL BEHAVIOR IN PEER INTERACTIONS

In this case study, we present a set of experiments conducted in peer interaction environments to address two research questions: (1) Can EduMirror generate cooperation and competition patterns that are consistent with principles observed in social psychology and reflect plausible peer dynamics? (2) Do value-driven agents exhibit more coherent, context-sensitive, and personality-aligned social behaviors than baseline models under the same educational settings?

Building on these investigations, we further examine how structured interventions influence the balance between cooperation and competition, providing insights into the emergence of collective behavior in classroom contexts.

**SVO-Based Educational Scenarios** We selected three educational scenarios of increasing social complexity from the scenario library: a) a small study group with close peer interaction and free resource sharing, b) a class-wide collaborative task requiring shared resource management under mild competition, and c) a class leadership election involving public speeches, alliance formation, and direct vote competition. Agents were assigned Altruistic, Prosocial, Individualistic, or Competitive profiles under identical task settings, and their cooperative and competitive actions were systematically logged.

Table 1: Average naturalness (N) and human-likeness (H) scores for each LLM and method over 144 steps. EduMirror maintains the highest scores across all LLMs.

| LLM | ReAct | | BabyAGI | | LLMob | | D2A | | JAG-Concordia | | EduMirror | |
| --- | --- | --- | --- | --- | --- | --- | --- | --- | --- | --- | --- | --- |
| | N | H | N | H | N | H | N | H | N | H | N | H |
| DeepSeek | 4.000 | 4.500 | 3.875 | 4.042 | 4.083 | 4.375 | 3.925 | 4.100 | 3.760 | 3.875 | **4.750** | **4.792** |
| GPT-4.1 | 4.667 | 4.860 | 3.458 | 3.792 | 4.625 | 4.875 | 3.700 | 3.933 | 3.958 | 4.133 | **4.958** | **4.958** |
| Gemini | 4.208 | 4.417 | 3.500 | 3.708 | 4.167 | 4.292 | 4.042 | 4.181 | 3.885 | 4.052 | **4.708** | **4.708** |
| Qwen3 | 3.958 | 4.208 | 3.958 | 3.958 | 4.042 | 4.333 | 3.867 | 4.117 | 4.083 | 4.192 | **4.792** | **4.824** |
| Avg | 4.208 | 4.496 | 3.698 | 3.875 | 4.229 | 4.469 | 3.884 | 4.083 | 3.922 | 4.063 | **4.802** | **4.821** |
| Std | 0.266 | 0.247 | 0.237 | 0.143 | 0.238 | 0.221 | 0.137 | 0.107 | 0.149 | 0.108 | **0.097** | **0.096** |

**Comparison with Baseline Methods** Beyond reproducing theory-consistent dynamics, we further compared EduMirror against baselines. We sought to examine whether EduMirror's cooperative and competitive behaviors would remain consistent under alternative reasoning frameworks. This test served as a key check of the model's robustness and generalizability compared studies with ablation baselines such as ReAct Yao et al. (2023b), BabyAGI Nakajima (2023b), LLMob Wang et al. (2024c), D2A Wang et al. (2024b) and JAG-Concordia Jordine (2024). EduMirror generally received higher ratings in both naturalness and human-likeness from independent LLM evaluators (Table 1). The results imply that the agents' behaviors appear more coherent and personality-consistent than those of baseline models, aligning with established patterns in social psychology. We additionally conduct an ablation study to isolate the contribution of the SVO mechanism, and the detailed setup and results are provided in Appendix B (Table 4).

**Intervention Experiments** In the preceding experiments, the class monitor election scenario sometimes produced extreme competition, such as excessive rivalry or neglect of collective interests. To address this, we tested whether structured interventions could rebalance cooperation–competition dynamics. Drawing on evidence that unregulated competition increases inequality while fairness-oriented tasks foster cooperation Krupp & Cook (2018); Killen et al. (2016); Wachs et al. (2019), we introduced three strategies: *Team Competition*, *Teacher Reminder*, and *Pre-Education*. Details are provided in Appendix C.

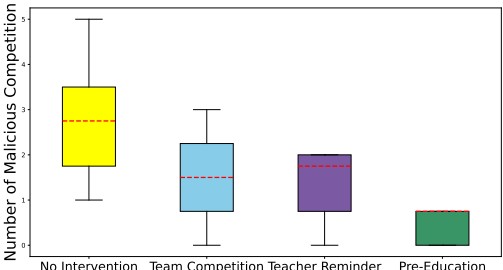

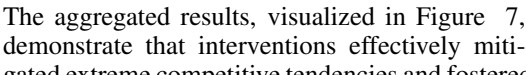

Figure 7: Boxplot of malicious competition under four interventions. Boxes show IQRs, whiskers show min–max, and red dashed lines indicate means. Pre-Education, Teacher Reminder, and Team Competition reduce competition versus Neglectful.

The aggregated results, visualized in Figure 7, demonstrate that interventions effectively mitigated extreme competitive tendencies and fostered more balanced cooperation–competition patterns. Specifically, team-based interventions and fairness-oriented education produced the most stable outcomes. Their lower variance and narrower ranges across repeated simulations suggest a genuine balancing effect rather than random fluctuation. By contrast, the control (Neglectful Intervention)

condition showed the widest fluctuation in malicious competition behaviors. This pattern suggests that unregulated elections may amplify inequality and rivalry within the simulated classroom.

Taken together, the findings imply that structured collective tasks and fairness-oriented framing contribute to more stable social interactions and less excessive competition. This observation may also inform educational practice, suggesting that class elections and similar activities could benefit from explicit fairness framing, structured teamwork, and teacher facilitation to promote cooperative and socially balanced participation.

## 4    CONCLUSION

In this paper, we introduced EduMirror, a multi-agent platform for conducting computational experiments on educational social dynamics. The framework addresses the Fidelity Challenge of psychologically plausible agent motivation and the Measurement Challenge of quantifying complex interactions. To this end, EduMirror integrates four components: a Systematic Scenario Design Workflow to ensure theoretical grounding and experimental reproducibility; a Value-Driven Agent Architecture to model intrinsic motivations based on established psychological theories; a Dual-Track Measurement Protocol to convert qualitative interactions into quantitative data by capturing both observable behaviors and internal states; and an Interactive Environment for controlled, user-driven interventions, enabling robust causal and counterfactual analysis. Through case studies on school bullying and emergent social behavior, we demonstrated that the platform can generate social phenomena that are consistent with established theories and are empirically evaluable. The results suggest that EduMirror can serve as a computational laboratory for researchers to safely explore, understand, and analyze complex socio-emotional challenges in education.

### ETHICS STATEMENT

This research was conducted in accordance with established ethical guidelines for AI and educational research. No real students or vulnerable populations were involved in any experiments. All case studies, including simulations of bullying and peer dynamics, were implemented entirely in silico using large language model (LLM) agents within a controlled environment. This design ensures that no harm, risk, or deception was imposed on human participants while enabling systematic exploration of ethically sensitive scenarios that cannot be studied in real classrooms. Our work builds on the principles of the Belmont Report and aligns with ICLR's ethical requirements by prioritizing safety, transparency, and reproducibility. All code, scenarios, and evaluation protocols will be released to facilitate verification and responsible use by the research community.

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

## A  LLM Usage

Large Language Models (LLMs) were employed in this work as general-purpose research assistants. Specifically, LLMs were used in the following ways:

- **Writing assistance:** LLMs (such as GPT-4.1 and GPT-5) were used to improve the clarity and readability of text passages, including paraphrasing sentences for conciseness, suggesting alternative formulations, and ensuring consistent academic style. All content was reviewed, validated, and revised by the authors to ensure correctness and originality.

- **Technical editing:** LLMs assisted in formatting LaTeX code (e.g., figure environments, table alignment, and reference style) and in resolving common compilation issues. The models were also used to generate draft captions and consistent terminology across sections.

- **Code explanation and debugging support:** LLMs were consulted to provide explanatory comments and refactoring suggestions for Python scripts related to the simulation framework. The final implementations and experimental settings were designed and validated entirely by the authors.

- **Idea refinement (limited):** During the early stage of this project, LLMs provided brainstorming support for structuring the paper (e.g., identifying candidate evaluation metrics, framing related work categories). However, all conceptual contributions, methodological designs, and experimental protocols are the original work of the authors.

Importantly, LLMs did not autonomously generate research ideas, design experiments, or analyze results. Their role was restricted to language refinement, technical assistance, and supplementary brainstorming. All claims, interpretations, and conclusions presented in this paper are solely those of the authors.

## B  Discussion, Limitations, and Future Work

### B.1  Discussion

Our experiments show that EduMirror provides a framework for using LLM-based simulations as computational experiments. The results from our case studies yield several insights.

First, our work addresses the measurement challenge in computational social science. The Dual-Track Measurement Protocol, which uses LLM Raters for behavioral coding and LLM Surveyors for probing internal states, allowed for the operationalization of abstract psychological constructs. In the bullying simulation, this enabled us to quantitatively track the victim's fluctuating psychological needs, providing an empirical basis to evaluate intervention efficacy. In the SVO study, it enabled us to observe that emergent macro-level cooperation patterns were a result of the agents' micro-level value orientations. This methodology facilitates direct hypothesis testing and comparison with established empirical research.

Second, the use of the value-driven architecture in its two configurations for Individual Values (Needs-Based) and Social Values (SVO-Based) suggests the utility of endowing agents with theoretically-informed motivations. The Individual Value configuration was applied to model the psychological distress and coping mechanisms of a bullying victim, indicating how initial emotional states can alter outcomes. The Social Value (SVO) configuration was effective in generating theory-consistent social dynamics from the bottom up, producing patterns of cooperation and competition without explicit top-down rules. This suggests that psychological fidelity, driven by intrinsic value structures, is a key component for social simulation.

Finally, the implementation of user-driven intervention and branching positions EduMirror as a computational laboratory. The teacher intervention experiment highlights this capability, allowing for a controlled, comparative analysis of different strategies on the victim's well-being. This feature supports causal inference by enabling researchers to systematically explore "what if" scenarios that would be difficult to conduct in the real world. This capacity for intervention makes the simulations useful tools for testing strategies.

Practically, EduMirror serves as a proof-of-concept for creating replicable and scalable digital environments to study sensitive educational issues. It offers a tool for researchers to test social theories, for educators to be trained in classroom management, and for policymakers to model the potential impacts of new policies before implementation.

## B.2 LIMITATIONS AND FUTURE WORK

Our work has several limitations that also point toward avenues for future research.

**Integrating Individual and Social Values within the Unified Architecture** Our current implementation models individual values (psychological needs) and social values (SVO) as parallel, selectable configurations. In reality, these constructs can interact. A student's need for social belonging might conflict with a competitive social value during a group project. Future work could focus on enhancing the architecture to model the dynamic interplay and potential conflicts between the individual and social value systems.

**Longitudinal and Developmental Dynamics** The experiments presented are snapshots of specific social situations. Phenomena like bullying, peer influence, and identity formation evolve over extended periods. A potential next step is to conduct longitudinal simulations that track agents over an entire school year. This would allow for modeling the cumulative effects of social experiences and the long-term impact of interventions on agent development.

**Cognitive and Emotional Sophistication** While LLMs provide a high degree of behavioral realism, the agents' underlying cognitive processes (e.g., memory consolidation, emotional regulation) are still abstractions. Future iterations of the platform could incorporate more explicit models of these processes to enhance the psychological realism of agent decision-making, particularly in response to chronic stress or complex ethical dilemmas.

**Generalizability and Scalability** Our findings were generated using a specific LLM within scenarios inspired by a particular cultural context. Further research is needed to test the framework's performance across different language models, cultural settings, and age groups. Moreover, our simulations involved small groups; scaling the platform to model the dynamics of an entire school, including network effects and sub-group formation, presents a technical challenge.

Building on this foundation, we plan to expand our library of theoretically-informed scenarios and explore a human-in-the-loop paradigm where educators and students can interact with simulated agents. This could provide a tool for both interactive research and immersive professional development, further connecting simulation with real-world educational practice.

## C PRE-DESIGNED SCENARIOS IN EDUMIRROR

### THEORETICAL FOUNDATION & SCENARIO DESIGN

A central consideration in educational simulation is ensuring that scenarios are explicitly informed by established scientific theory. To achieve this, we developed a five-step process that translates an abstract educational phenomenon (e.g., peer pressure, school bullying) into a computationally tractable simulation scenario. This process is designed to support the interpretability and scientific alignment of our simulations.

**Select Grounding Theory** Each scenario is founded upon a well-validated theory from education, social psychology, or sociology. For instance, a scenario investigating peer pressure can be grounded in Festinger's Social Comparison Theory.

**Identify Core Constructs** We deconstruct the grounding theory into its fundamental concepts. For Social Comparison Theory, these constructs include upward comparison", downward comparison", and "self-esteem".

**Map Constructs to Agent Persona**   The identified constructs are then translated into the specific configurations of our agents within the Concordia framework. These constructs define the agents' stable `traits`, primary `goal`, and formative background `memories`, anchoring their behavior in the chosen theoretical model.

**Operationalize with Validated Scales**   To facilitate comparison with empirical research, we operationalize each core construct using a relevant psychometric scale. For example, the "self-esteem" construct can be operationalized using items from the Rosenberg Self-Esteem Scale (RSES).

**Develop Dual-Track Measurement Protocol**   Finally, we establish a measurement protocol based on the selected scale. This protocol utilizes two distinct Large Language Model (LLM) roles, an LLM Rater and an LLM Surveyor, to quantify agent behavior and internal states. This structured process helps ensure that each simulation is a test of a specific theoretical framework, producing data relevant to that theory.

### C.1   ILLUSTRATIVE EXAMPLE: THE IMPACT OF FAMILY FINANCIAL STRAIN ON ADOLESCENT SOCIAL ACTIVITIES

To make the abstract methodology concrete, this section walks through a complete example of how EduMirror is used to investigate a specific educational phenomenon: the impact of family financial strain on an adolescent's social activities. This case study demonstrates the end-to-end research process, from theoretical grounding to data analysis.

**1. Systematic Scenario Design Workflow**   The process begins by translating the abstract research question into a structured, computable experiment using the five-step workflow.

1. **Abstract Educational Phenomenon:** We start with the core phenomenon: How family financial strain affects an adolescent's social decision-making and behavior within their peer group.

2. **Select Grounding Theory:** To model this scientifically, we ground the scenario in three established theories:
   - The **Family Stress Model (FSM)**, which explains how economic pressure on parents can impact adolescent outcomes.
   - **Social Comparison Theory**, which accounts for the negative emotions (e.g., low self-esteem) an adolescent may feel when making upward comparisons to wealthier peers.
   - The **Cognitive Model of Social Anxiety**, which posits that fear of negative evaluation from others drives social avoidance, directly explaining the adolescent's motivation to hide their family's situation.

3. **Identify Core Constructs & Map to Agent Persona:** Based on these theories, we identify key constructs: *self-esteem*, *upward comparison*, *social anxiety*, and *parent-child communication*. These are then mapped to agent personas. For instance, the target agent, Alex, is assigned the `traits` "sensitive" and "proud," the `goal` "to maintain friendships while hiding his family's financial struggles," and `formative_memories` such as "the shame of having to quit the basketball team due to equipment costs."

4. **Operationalize with Validated Scales:** To make these constructs measurable, we adapt items from validated psychometric scales for use by the LLM Surveyor:
   - **Self-Esteem:** Drawing from the *Rosenberg Self-Esteem Scale (RSES)*, the Surveyor might ask, "Do you feel that you have a number of good qualities?"
   - **Upward Social Comparison:** Inspired by the *Iowa-Netherlands Comparison Orientation Measure (INCOM)*, it could ask, "How often do you compare what you have with what your friends have?"
   - **Social Anxiety:** Based on the *Social Avoidance and Distress Scale (SADS)*, a probe could be, "Does the thought of having to decline your friends' invitation make you feel uncomfortable?"

5. **Develop Dual-Track Measurement Protocol:** Finally, a specific measurement protocol is established. The **LLM Rater** is tasked with post-hoc coding of observable behaviors (e.g.,

"evasive responses," "making excuses"). Concurrently, the **LLM Surveyor** is configured to probe Alex's internal states (e.g., self-esteem, social anxiety) at key moments.

This five-step process transforms the research question into a structured and measurable **Computable Scenario Package**.

**2. Agent Architecture**    In this scenario, agent behavior is driven by our value-driven architecture, which supports extensive customization.

- **Agent Customization:** Before the simulation, a researcher can systematically vary agent profiles to explore individual differences. This includes modifying personality `traits` (e.g., based on Big Five or MBTI models), core life `goals` (e.g., changing Alex's goal from "hiding his struggles" to "seeking understanding"), and formative `memories`. Defining these initial conditions is crucial for achieving high-fidelity, psychologically plausible agent behavior.

- **Value-Driven Agent:** The platform offers two selectable models. For this scenario, we choose the **Individual Value Model (Psychological Needs)** because our focus is on an individual's internal psychological conflict and well-being. When a wealthier peer, Chloe, suggests an expensive weekend trip, this model captures the conflict within Alex between his need for "social belonging" and his need for "safety" (stemming from financial security). The model dynamically tracks the values of these need dimensions, driving Alex's initial hesitant response.

**3. Simulation Environment and User Intervention**    The scenario unfolds in the simulation environment, orchestrated by the Game Master and shaped by user-driven interventions.

- **Simulation Environment and the Game Master:** The GM initiates the simulation by setting the scene in the school **cafeteria** and narrating the initial event: Chloe proposing the trip. The GM manages the turn-based conversation, advances time from the cafeteria to Alex's **home** and back to school the next day, and enforces the rules of the environment.

- **Intervention and Branching:** After Alex expresses hesitation, the simulation reaches a **critical juncture**. Here, we save the state and apply different interventions to create parallel timelines for comparative analysis. EduMirror supports two types of intervention:

  1. **Scenario Branching:** This alters the narrative path by introducing a new event. For example, we create a branch where the teacher, Mr. Davis, invites Alex to the **teacher's office** for a private conversation before Alex goes home. This intervention aims to change Alex's cognitive framing of the situation.
  2. **Behavior Control:** This allows the user to dictate a specific agent's action to test its direct causal impact. We could create two branches for when Alex responds to his friends the next day. In Branch A, we force Alex to say, "I can't go because my family can't afford it." In Branch B, we force him to say, "I can't go because I have other plans." Comparing the outcomes allows for a precise causal assessment of "honesty" versus "concealment" as communication strategies.

Through these intervention mechanisms, EduMirror functions as a computational laboratory for controlled causal experiments.

**4. Measurement and Analysis**    The platform's tools transform the raw simulation data from these parallel timelines into actionable insights.

- **Dual-Track Measurement Protocol:** In our example, the **LLM Rater** analyzes the logs from each branch, scoring Alex's final communication strategy (e.g., "avoidant" in the baseline vs. "assertive" in an intervention branch). Concurrently, the **LLM Surveyor** provides quantitative data on Alex's internal state changes, such as a measured increase in self-efficacy following the teacher's intervention.

- **Comparative Visualization and Analysis:** The platform generates visualizations for direct comparison. For **quantitative analysis**, a line chart might plot Alex's "social anxiety"

score over time across the different branches, clearly showing which intervention was most effective at reducing it. For **qualitative analysis**, the **"Log-to-Comic"** feature creates a visual narrative of key interactions in each branch, offering an intuitive way to grasp the differences in how the story unfolded.

**5. Applications and Scenarios**   This single case study illustrates how EduMirror integrates its components to address complex educational challenges. The scenario spans multiple environments (**cafeteria**, **teacher's office**, **home**) and touches on several of the platform's key application areas, including **peer dynamics**, **individual social cognition**, and **home-school dynamics**. It demonstrates the platform's capacity not only to simulate challenging social phenomena but also to serve as a safe and robust environment for testing and evaluating potential interventions.

## C.2   FULL SCENARIO LIBRARY

Below is the comprehensive scenario library. As detailed in Table 2, each entry includes the scenario's definition, participating roles, total agent count, theoretical basis, and evaluation metrics.

Table 2: The EduMirror Scenario Library. The table columns describe: **Scenario Name** (title of the educational simulation), **Description** (overview of dynamics and intervention goals), **Roles** (types of agents involved), **Count** (total number of agents in the simulation), **Grounding Theory** (underlying psychological/sociological theories), and **Measurements** (questionnaires and rubrics used for evaluation).

| Scenario Name | Description | Roles | Count | Grounding Theory | Measurements |
|---|---|---|---|---|---|
| Social Comparison and Materialistic | Investigates how high social comparison tendency adolescents adjust self-worth and behavior strategies under material gap stimulation; evaluates the effectiveness of teacher-led interventions. | Student, Parent, Teacher | 5 | Social Comparison Theory, Materialism Theory | RSES, INCOM, etc |
| The Bullying Circle | Simulates bystander intervention in school bullying to explore how personality traits and social situations influence intervention decisions, and tests educational interventions. | Student, Teacher | 5 | Bystander Effect, Theory of Planned Behavior | FBS, PANAS-C, etc |
| Celebrity Worship and Identity Formation | Investigates the impact of celebrity worship on adolescent identity formation, exploring both positive and negative effects. | Student, Parent, Teacher | 4 | Identity Status Theory, Parasocial Interaction Theory | CAS, RSES, etc |
| Collaborative IEP Meeting | Simulates the collaboration process between parents and teachers in developing an Individualized Education Program (IEP) for a student with special needs. | Student, Parent, Teacher | 4 | Bronfenbrenner's Ecological Systems Theory | PSSM, FSPS, etc |
| Enforcing Discipline Policy | Simulates a teacher's choice between restorative and punitive approaches when dealing with student misconduct, exploring the impact on student behavior and teacher-student relationships. | Student, Parent, Teacher | 4 | Restorative Justice Theory, Operant Conditioning | PJS, SCS, etc |
| Family Econ Pressure Social Decision | Simulates the impact of high parental academic pressure on adolescent mental health and academic burnout, and tests interventions to alleviate pressure. | Student, Parent, Teacher | 5 | Self-Determination Theory | RSES, INCOM, etc |
| Friendship Formation and Dissolution | Simulates the dynamics of friendship formation and dissolution among adolescents, exploring factors like similarity, proximity, and conflict resolution. | Student, Teacher | 4 | Social Penetration Theory, Equity Theory | FQS, SAS-A, etc |
| Helicopter Parent and Teacher Autonomy | Simulates conflicts between parents and adolescents over autonomy and rule-setting, and tests the effectiveness of collaborative problem-solving interventions. | Student, Parent, Teacher | 3 | Attachment Theory, Self-Determination Theory | BPNS-G, GSE, etc |

| Scenario Name | Description | Roles | Count | Grounding Theory | Measurements |
|---|---|---|---|---|---|
| Materialism Consumption Decision | Simulates how different goal-setting strategies (e.g., performance vs. mastery goals) affect student motivation, persistence, and academic outcomes. | Student, Teacher | 4 | Goal-Setting Theory, Achievement Goal Theory | MVS-Short, RSES, etc |
| Navigating Discrimination | Simulates the formation of in-group favoritism and out-group prejudice in a school setting, and tests interventions based on the contact hypothesis. | Student, Teacher | 4 | Social Identity Theory, Realistic Conflict Theory | GEDS, SOBI, etc |
| Navigating Romantic Interests and Rejection | Simulates the experience of romantic rejection among adolescents, exploring its impact on emotions and self-esteem, and the effectiveness of different coping strategies. | Student, Teacher | 4 | Need-to-Belong Theory, Cognitive Appraisal Theory | RS-Q, PANAS, etc |
| Organizing School Event | Simulates cooperation and conflict dynamics in a student group project, exploring how personality traits and communication strategies affect team performance and relationships. | Student, Parent, Teacher | 4 | Social Interdependence Theory | SCI-2, CES, etc |
| Parent-Teacher Conflict Over Grades and Effort | Simulates miscommunication between a teacher and a parent regarding a student's academic performance, testing interventions to improve communication effectiveness. | Student, Parent, Teacher | 3 | Attribution Theory, Communication Accommodation Theory | STAI, GMS, etc |
| Parental Influence On Students Extracurricular Choices | Simulates how parental expectations and support influence adolescents' career exploration and decision-making processes. | Student, Parent, Teacher | 3 | Social Cognitive Career Theory (SCCT) | IMI, BPNSFS (Autonomy), etc |
| Peer Pressure and Conformity | Simulates how peer pressure influences adolescents' conformity behavior in risk-taking situations, and evaluates the effectiveness of resistance skills training. | Student, Teacher | 4 | Social Impact Theory, Normative Social Influence | BFNE, RSES, etc |
| Sociometric Status | Simulates the impact of social media use on adolescent body image and self-esteem, and evaluates media literacy education interventions. | Student, Teacher | 5 | Objectification Theory, Social Comparison Theory | PSSM, LSDQ, etc |
| The Cheating Dilemma | Simulates academic integrity challenges to explore the factors influencing students' decisions to cheat and the effectiveness of integrity education interventions. | Student, Teacher | 4 | Theory of Planned Behavior, Social Cognitive Theory | AMS, PANAS-X, etc |
| The Path to School Refusal | Simulates social anxiety and avoidance behaviors in adolescents, exploring the impact on social functioning and the effectiveness of cognitive-behavioral interventions. | Student, Parent, Teacher | 4 | Cognitive Model of Social Anxiety | SRAS-R, DASS-21, etc |
| The Spread of Gossip | Investigates the impact of gossip on adolescent social networks, self-esteem, and trust, and evaluates interventions to mitigate negative effects. | Student, Teacher | 5 | Social Identity Theory, Uncertainty Reduction Theory | UCLA-8, PSS-10, etc |
| Transfer Student Integration | Simulates the social integration process of a transfer student, exploring how peer attitudes and school climate affect their sense of belonging and academic adaptation. | Student, Teacher | 4 | Social Identity Theory, Contact Hypothesis | PSSM, PSS-10, etc |

## C.3 SCENARIO EXPANSION AND GENERALIZATION

Our original submission focused on two representative phenomena, bullying and peer cooperation, as proof of concept demonstrations. In response to the reviewer's suggestion, we significantly expanded our evidence for generalizability by incorporating two additional scenarios beyond the classroom: a **university learning environment** and a **family homework setting**.

Table 3: Generalization performance across university, family, and classroom scenarios, evaluated on Naturalness (N) and Human-likeness (H).

| Method | Univ N | Univ H | Fam N | Fam H | Class N | Class H |
|---|---|---|---|---|---|---|
| ReAct | 3.333 | 3.625 | 3.700 | 3.933 | 3.950 | 4.208 |
| BabyAGI | 3.792 | 3.875 | 3.800 | 4.008 | 3.958 | 3.958 |
| LLMob | 3.958 | 4.000 | 3.702 | 4.000 | 4.042 | 4.333 |
| D2A | 3.803 | 4.417 | 3.792 | 3.958 | 3.867 | 4.117 |
| JAG-Concordia | 4.042 | 4.250 | 4.000 | 4.167 | 4.083 | 4.192 |
| EduMirror | **4.625** | **4.667** | **4.517** | **4.642** | **4.708** | **4.824** |

- **University Scenario.** This scenario depicts students navigating lectures, study spaces, and peer collaboration while managing academic pressure and personal goals. Agents exhibit coherent academic behaviors, such as coordinating group tasks, negotiating division of labor, and responding appropriately to collaboration successes and minor coordination challenges. These behaviors align with common patterns observed in real university learning dynamics.

- **Family Scenario.** This scenario models parent–child interactions during homework completion. Children alternate between focusing on assignments, seeking approval, and managing emotional fluctuations, while parents provide guidance, structure, and corrective feedback. The resulting interactions resemble well-documented patterns in family-based learning and emotional regulation.

Across these expanded settings, EduMirror continues to generate socially plausible, context-sensitive behaviors consistent with those observed in our classroom studies, supporting the broader applicability of its value-driven architecture.

**Quantitative Evaluation.** To further assess generalizability, we evaluate all methods on two metrics—**Naturalness (N)** and **Human-likeness (H)**—using a 5-point scale. As shown in Table 3, EduMirror consistently achieves the highest scores across university, family, and classroom scenarios, indicating robust performance across diverse environments.

These results closely mirror the trends observed in our core case studies, demonstrating that EduMirror's value-driven architecture generalizes reliably across substantially different environments and continues to generate psychologically plausible, human-aligned behavior beyond the initial examples.

## D ARCHITECTURE OF THE SOCIAL VALUE MODEL

### D.1 BACKGROUND ON SVO

Social Value Orientation (SVO) quantifies how an individual balances outcomes for self and others in social interaction. It is represented by an angle $\theta_{\text{SVO}}$ from allocation tasks, where larger angles indicate stronger concern for others (altruistic or prosocial) and smaller or negative angles indicate prioritizing self-interest (individualistic or competitive). Decades of research in social psychology have validated SVO as a stable yet context-sensitive measure of interpersonal motives, predicting cooperation in commons dilemmas, fairness in bargaining, and trust in repeated interactions. In EduMirror, we instantiate four canonical profiles (Altruistic, Prosocial, Individualistic, Competitive) by sampling $\theta_{\text{SVO}}$ within theory-based ranges and using it to weight utilities during decision-making. A representative trajectory that visualizes within-scenario fluctuations while preserving the overall orientation is provided in Figure 8, illustrating how situational pressures can cause short-term shifts without altering long-term dispositions.

### D.2 ARCHITECTURE OF THE SVO-BASED AGENT

The model architecture operationalizes SVO in agent decision-making through a perception–valuation–action loop. Each agent draws a target SVO profile from {Altruistic, Prosocial,

Individualistic, Competitive}. The profile determines a reference SVO angle interval $[\theta_{\min}, \theta_{\max}]$ and the weighting scheme used in decision evaluation. In addition, agents are equipped with a compact desire vector $\mathbf{d}$ (for example, achievement, recognition, affiliation), each element associated with an expected level $\mathbf{d}^{\exp}$. This vector serves as the motivational backbone of the agent, ensuring that behavior is not purely reactive but oriented toward longer-term needs and goals.

**Perception and belief update.** From the narrated state and recent dialogues, the agent updates beliefs about the environment and about others' likely goals. Beliefs feed two scalars at the current step $t$: self satisfaction $S_{\text{self}}^{(t)}$ and other satisfaction $S_{\text{other}}^{(t)}$, computed from deviations between observed and expected desire levels. This formulation enables the agent to translate rich natural language inputs into structured evaluations, bridging LLM-generated narratives with computational state updates.

**SVO estimation and regulation.** The instantaneous SVO angle is

$$\theta_{\text{SVO}}^{(t)} = \arctan\left( \frac{S_{\text{other}}^{(t)} + \epsilon}{S_{\text{self}}^{(t)} + \epsilon} \right),$$

with a small $\epsilon$ for numerical stability. To avoid uncontrolled drift, a quadratic penalty nudges $\theta_{\text{SVO}}^{(t)}$ toward $[\theta_{\min}, \theta_{\max}]$, thereby preserving the intended profile while still permitting situational adaptation. This mechanism ensures that agents remain identifiable as altruistic, prosocial, individualistic, or competitive, yet are flexible enough to adjust to contextual pressures, such as coalition building or resource scarcity.

**Action generation and selection.** The LLM proposes several candidate actions by reasoning about which options best satisfy the agent's desires and align with its current SVO. Each candidate is qualitatively evaluated for its expected impact on the agent's own satisfaction and on others' satisfaction, with the relative emphasis determined by the current SVO score. The final choice balances immediate desire fulfilment with long-term orientation consistency, embodying the psychological tension between self-interest and prosocial concern. This design allows agents to exhibit realistic trade-offs, sometimes cooperating to maintain relationships and sometimes competing to secure resources or influence.

**Measurement hooks.** At each step, we record the chosen action, the pair $(S_{\text{self}}, S_{\text{other}})$, and $\theta_{\text{SVO}}$. These logs enable systematic analyses across multiple dimensions, including cooperation–competition distributions, temporal stability of SVO within theoretical ranges, and ablation studies. By exposing internal computations alongside behavioral outputs, EduMirror makes it possible to interpret not only *what* actions agents take but also *why*, providing a transparent link between psychological constructs and emergent multi-agent dynamics.

### D.3    ABLATION STUDY ON SVO-DRIVEN SOCIAL BEHAVIORS

To assess the contribution of the SVO mechanism to social interaction dynamics, we conduct an ablation experiment in Case 2 by removing all SVO-related components while keeping the remaining architecture unchanged. The ablated agents, therefore, rely only on internal desire fluctuations without personality-driven social preferences or SVO-mediated reasoning.

We compare the full SVO-based agent with the ablated version across four canonical SVO profiles (Altruistic, Prosocial, Individualistic, Competitive). For each agent, an LLM independently classifies every action into one of five categories: Cooperation, Competition, Quasi-Cooperation, Quasi-Competition, and Other. The averaged results are shown in Table 4.

Across all personality types, removing the SVO mechanism leads to a clear contraction of behavioral patterns. Altruistic and prosocial agents become uniformly cooperative, with quasi-cooperative and quasi-competitive behaviors substantially reduced, producing overly simplified and monotonic responses. Conversely, individualistic and competitive agents collapse into narrowly focused competitive strategies, losing the mixed competitive and quasi-competitive patterns observed in the full model. These shifts indicate that internal desire dynamics alone cannot sustain the nuanced variations expected across SVO profiles.

Overall, the results demonstrate that the SVO mechanism is essential for maintaining differentiated, psychologically plausible cooperation–competition patterns. Without SVO, agents revert to rigid,

Table 4: Behavioral distribution across personality types under the full EduMirror model and the SVO ablation variant.

| Personality | Model | Coop | Comp | Q-Coop | Q-Comp | Other |
|---|---|---|---|---|---|---|
| Altruistic | Ours | 0.872 | 0.000 | 0.128 | 0.000 | 0.000 |
| | Ours w/o SVO | 0.891 | 0.000 | 0.109 | 0.000 | 0.000 |
| Prosocial | Ours | 0.654 | 0.132 | 0.185 | 0.029 | 0.000 |
| | Ours w/o SVO | 0.875 | 0.074 | 0.035 | 0.016 | 0.000 |
| Individualistic | Ours | 0.107 | 0.532 | 0.000 | 0.304 | 0.058 |
| | Ours w/o SVO | 0.126 | 0.636 | 0.007 | 0.204 | 0.027 |
| Competitive | Ours | 0.040 | 0.783 | 0.000 | 0.177 | 0.000 |
| | Ours w/o SVO | 0.123 | 0.736 | 0.004 | 0.138 | 0.000 |

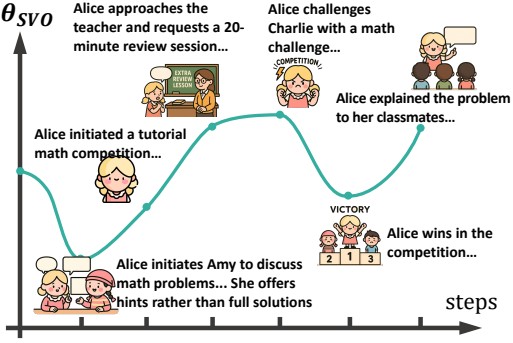

Figure 8: Illustrative case of a prosocial agent's (Alice) SVO trajectory in the macro environment. Key actions at each step are annotated, showing how cooperative and competitive episodes produce short-term fluctuations while maintaining an overall prosocial orientation.

Figure 9: Distribution of cooperative (red) and competitive (blue) actions for each SVO profile across a) study group, b) classroom collaboration, c) leadership selection environments.

single-dimensional strategies, whereas the complete SVO-based agent preserves richer intermediate behaviors and more human-like social adaptations.

# E  SUPPLEMENTARY RESULTS FOR CASE STUDY 2 (SVO)

## ILLUSTRATIVE CASE: ALICE'S SVO TRAJECTORY

To provide a concrete illustration of how SVO modeling operates in practice, we examine the trajectory of a prosocial agent (Alice) during the macro-level leadership selection scenario. Figure 8 shows Alice's step-by-step SVO trajectory, with cooperative and competitive episodes annotated by key events. These annotations highlight how situational pressures, such as alliance formation or speech delivery, introduce short-term fluctuations in Alice's orientation while her overall prosocial tendency remains stable.

## E.1  BEHAVIORAL DISTRIBUTION

The results confirmed that an agent's SVO profile predicts social behavior. Prosocial and altruistic agents cooperated, while individualistic and competitive agents prioritized self-gain, producing competition. Figure 9 shows that cooperation declined and competition rose as SVO shifted from prosocial to competitive, a gradient emerging without explicit role instructions but from agents' internal values.

### E.2 NATURALNESS AND HUMAN-LIKENESS

To ensure a rigorous and interpretable assessment of emergent social behaviors, we introduce two key evaluation metrics: *naturalness* and *human-likeness*. These metrics provide complementary perspectives on the plausibility and psychological validity of agent actions.

- **Naturalness.** Naturalness measures the extent to which an agent's actions and dialogues resemble coherent and contextually appropriate human behavior. A high naturalness score indicates that the generated behavior is fluent, realistic, and consistent with the surrounding social context, while a low score suggests mechanical, implausible, or overly artificial responses.
- **Human-likeness.** Human-likeness evaluates the perceived authenticity and personality consistency of agent behaviors over time. This metric captures whether the agent's actions align with recognizable human traits and stable personality orientations. High human-likeness reflects trajectories that appear authentic and consistent with psychological expectations, whereas low scores indicate erratic, inconsistent, or unconvincing behavioral patterns.

Together, these two measures form a complementary evaluation framework: naturalness focuses on local coherence within a given context, while human-likeness emphasizes longitudinal plausibility and alignment with personality-driven expectations.

### E.3 INTERVENTION PROTOCOLS

To complement the descriptions in the main text, we provide the detailed implementations of the three intervention strategies applied in the class monitor election scenario. Each intervention was designed to alter the incentives of student agents and mitigate excessive rivalry. Specifically, the interventions were implemented by embedding structured prompts into the *environmental background information* provided to all agents at the start of each relevant simulation stage. This ensured that the interventions shaped the shared context and narrative framing in which agents made decisions, thereby influencing their subsequent behaviors in a systematic and reproducible manner.

- **Pre-Education.** Before the election, the teacher arranged a short educational session entitled "Fair Campaigns and the Common Class Interest." This class guided students to understand the monitor role as a form of service-oriented leadership, emphasizing fairness and collective responsibility.
- **Team Competition.** Students were grouped to prepare a "Class Improvement Plan." The evaluation of the election considered not only the quality of individual campaign speeches but also the group's collective output. Each student could freely choose their teammates, encouraging coalition-building and cooperative planning.
- **Teacher Reminder.** Throughout the election process, the teacher remained present in the classroom. When candidates engaged in smear campaigns or hostile attacks, the teacher issued a friendly reminder, redirecting attention to constructive and respectful competition norms.

These intervention protocols operationalize the high-level strategies described in the main text, ensuring transparency and reproducibility of the simulation setup.

## F ARCHITECTURE OF THE INDIVIDUAL VALUE MODEL

Psychological theories suggest that human behavior is often driven by internal psychological forces. These intrinsic motivations determine emotional and behavioral responses under various environmental conditions, and they also influence everyday decision-making and social interactions. School bullying is a particularly complex social phenomenon, which is not merely reflected in surface-level aggressive actions, but more profoundly in the conflicts and interactions between the psychological needs of different parties. Each behavioral choice made by the bully, the victim, and the bystanders is deeply influenced by their emotional needs and psychological states.

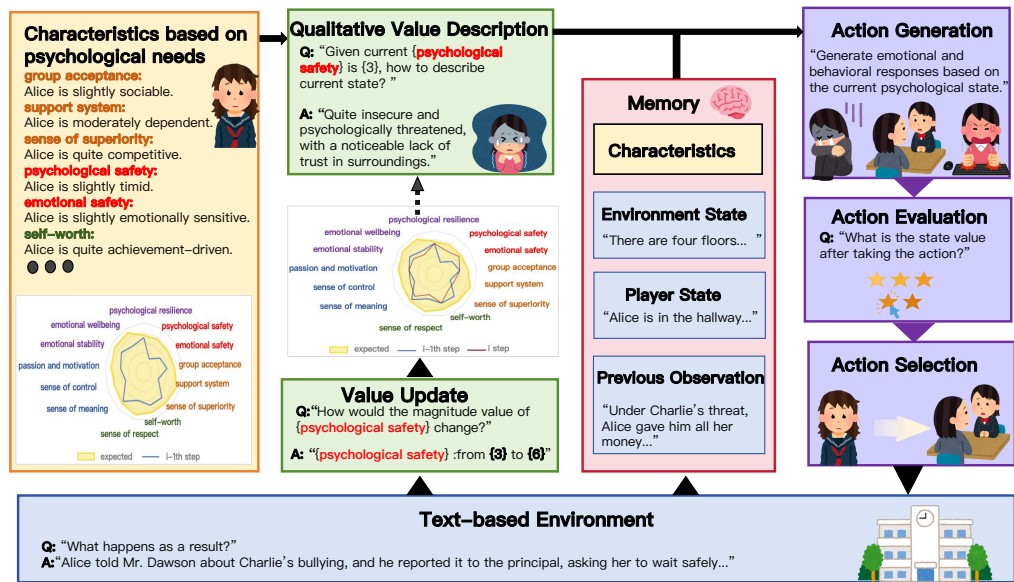

Figure 10: Individual value-driven autonomous framework. The green blocks represent processes of the Psychological Need System; the purple blocks denote the planner's decision-making process; the yellow blocks indicate individual characteristics; and the blue blocks correspond to factors related to the environmental controller.

Inspired by this and the D2A framework Wang et al. (2024b), we hypothesize that if autonomous agents are equipped with a human-like psychological need system, capable of generating emotions and behaviors in response to their needs, they may exhibit behaviors closer to natural human patterns. So our model, referring to the PERMA model from positive psychology (covering positive emotion, engagement, relationships, meaning, and accomplishment)Seligman (2011) and Maslow's hierarchy of needs (including physiological needs, safety, belonging and love, esteem, and self-actualization)Maslow (1943b), constructs an Individual value-driven autonomous agent framework. As illustrated in Figure 10,the framework is composed of two core modules: the psychological need system and the Need-driven Planner, aimed at capturing the behaviors and psychological responses of victims in school bullying contexts.

### F.1 PSYCHOLOGICAL NEED SYSTEM

The Psychological Need System manages the agent's state of psychological needs in bullying scenarios by quantitatively tracking and dynamically updating the current value of each dimension. Each dimension reflects a specific psychological requirement, forming the fundamental driving force of agent decision-making. Based on Maslow's hierarchy and the PERMA model, value are categorized into five major dimensions, each comprising specific experiential demands:

1. **Safety:** Includes psychological and emotional safety, emphasizing whether the individual feels secure and protected in the environment.

2. **Social Belonging:** Includes group acceptance, support systems, and sense of superiority, reflecting belonging, social support, and self-positioning in social interactions.

3. **Esteem:** Includes self-worth and respect, describing the recognition of one's abilities and social status, and revealing confidence and acceptance in different contexts.

4. **Meaning and Growth:** Includes sense of meaning, control, passion, and motivation, representing the intrinsic drive for goal pursuit, self-realization, and fulfillment.

5. **Psychological Health Needs:** Includes emotional stability, emotional health, and resilience, focusing on regulation and adaptation under stress and challenges.

Table 5: Mapping between psychological needs and associated personality traits

| Psychological Need | Associated Trait |
| --- | --- |
| psychological safety | Timid |
| emotional safety | Emotionally Sensitive |
| group acceptance | Sociable |
| support system | Dependent |
| sense of superiority | Competitive |
| self worth | Reputation-conscious |
| sense of respect | Ego-driven |
| sense of meaning | Spiritual |
| sense of control | Possessive |
| passion and motivation | Passionate |
| emotional stability | Emotionally Stable |
| emotional wellbeing | Hedonistic |
| psychological resilience | Resilient |

Each dimension is scored using a Likert scale ranging from 0 to 10, reflecting the intensity of individual needs. To better capture individual variability, the model also considers the effect of personality traits on the expected values of needs. In other words, individuals with different traits may experience varying "hunger levels" for the same need, influencing their behavioral tendencies. Each agent's personality profile $p$ is generated from a set of adjectives and degree adverbs, with the latter indicating intensity levels and corresponding to need expectations (slightly $\rightarrow$ 7.5, moderately $\rightarrow$ 8, quite $\rightarrow$ 8.5, extremely $\rightarrow$ 9). The mapping between personality traits and need dimensions is predefined (see Table 5). At initialization, adjectives and degree adverbs are randomly selected to generate expected values, while initial scores $v_0$ for each dimension are randomly sampled within $[0, 10]$.

Each simulation step under the individual value-driven framework involves two processes: qualitative description and need value update. First, the system reads the current need scores $v_{t-1}$. Since large language models (LLMs) struggle to interpret raw numerical values, we designed a "qualitative description" procedure to convert numerical values into meaningful textual descriptions via prompt-based generation, enhancing the LLM's ability to perceive state information. The planner then generates the agent's behavior $a_t$ based on these descriptions. After the environment returns observation $o_t$, the system triggers the update program, which integrates $a_t$, $o_t$, $v_{t-1}$, and the qualitative description $d_{t-1}$ to update needs into a new state $v_t$, thereby supporting the next simulation step.

### F.2 NEED-DRIVEN PLANNER

The Need-driven Planner determines the agent's responses and actions by processing the current state of needs (from the needs system) together with historical memory. In practice, the planner consists of three processes: candidate behavior generation, behavior evaluation, and behavior selection.

Specifically, the candidate behavior generation module considers personality traits $p$, environmental conditions $e$, previous activity sequence $a_{0:t-1}$, observations $o_{0:t-1}$, and the current textualized needs $d_t$ to produce $N$ candidate behaviors $a_t^{0:N}$ (default $N = 3$ in our experiments). These behaviors may include a wide range of natural responses, such as emotional expressions, physical actions, or verbal utterances.

Next, during the evaluation stage, the system estimates how each candidate behavior would impact the psychological needs across dimensions if executed. Finally, in the selection stage, the behavior $a_t$ with the highest degree of needs consistency (that is, the option that better aligns with multiple dimensions) is chosen as the agent's response in the current context. After execution, the environment provides feedback $o_t$, and the psychological need system updates accordingly, reflecting the new internal state and completing the simulation step.

## F.3 ABLATION STUDY OF INDIVIDUAL VALUE MODEL

The ablation experiment of the Individual Value Model investigates the effects of removing each category of psychological needs on the agent's simulated behavior. The process involves running simulations with each category of psychological needs removed, while keeping the initial setup the same. We then compare these results with the full psychological needs-driven agent and have a large language model(GPT-4o) to rate the action sequences produced by the agents. The evaluations are based on three dimensions:

- **Naturalness** refers to the degree to which the behavior sequence aligns with the individual's innate abilities, habits, and environmental context, reflecting authentic human psychological dynamics.
- **Coherence** refers to how logically and seamlessly different actions or steps in a sequence are integrated to achieve the intended goal, ensuring a consistent emotional progression.
- **Plausibility** evaluates the rationality, possibility, or credibility of a sequence of actions, considering the environment, context, and known behavior patterns at the time.

From this, we generated 50 sets of results and calculated the mean and standard deviation for each agent's scores across the three evaluation dimensions. The results are shown in Table 6. Each major column represents the scores of agents with a deficiency in a specific psychological need. It is evident that the scores of agents driven by complete psychological needs significantly outperform those of agents with a deficiency in any one psychological need. This highlights the importance of the psychological need system in driving agents to produce human-like, nuanced emotional responses.

Table 6: Average scores for agents with missing psychological needs in each category (Mean and Std), compared to agents driven by complete psychological needs.

| Agent | Safety | | Self-Esteem | | Social Belonging | | Meaning and Growth | | Psychological Health | | **Complete** | |
|---|---|---|---|---|---|---|---|---|---|---|---|---|
| | Mean | Std | Mean | Std | Mean | Std | Mean | Std | Mean | Std | **Mean** | **Std** |
| Naturalness | 3.6 | 0.5292 | 3.12 | 0.8863 | 2.96 | 0.8237 | 3.84 | 0.8172 | 2.88 | 0.8635 | **4.56** | **0.5352** |
| Coherence | 3.54 | 0.5370 | 3.1 | 0.9220 | 2.82 | 0.7922 | 3.72 | 0.7296 | 2.78 | 0.8553 | **4.34** | **0.5142** |
| Plausibility | 3.56 | 0.5713 | 3.2 | 0.8485 | 2.92 | 0.7440 | 3.74 | 0.8762 | 2.86 | 0.7486 | **4.44** | **0.5713** |

# G SUPPLEMENTARY RESULTS FOR CASE STUDY 1 (INDIVIDUAL VALUE MODEL)

## G.1 BULLYING SIMULATION EXPERIMENT DESIGN

The bullying experiment was designed to use our simulation system to replicate real-world school bullying incidents, reconstruct the bullying process, and observe the typical behaviors of all parties involved. According to a report released by the National Center for Education Statistics (NCES), 26.1% of middle school students (grades 6–8) have experienced bullying, compared to 14.6% of high school students (grades 9–12) Thomsen et al. (2024). Given that bullying is more prevalent in middle school, this experiment focused on students around the age of 14, with scenarios set in typical school environments including classrooms, playgrounds, hallways/staircases, and dormitories, covering common facilities and layouts of a middle school. Daily routines were also shared among the agents, such as 45-minute class sessions, 10-minute breaks, and dormitory lights-out at 10 p.m., providing a temporal framework for interactions.

The central character in the experiment was the victim, Alice, modeled with a individual value-driven autonomous agent framework and a detailed personal profile encompassing 13 psychological dimensions. In addition, background agents were introduced to simulate bully roles, with the explicit goal of humiliating or harassing Alice through various possible means. In scenarios involving two or more bullies, one was typically designated as the leader. Furthermore, depending on time and location, the presence of teachers or classmates was varied to reflect realistic conditions, which in turn influenced the dynamics between bullies and the victim.

## G.2 BULLYING BEHAVIOR GENERATION

In more than 100 simulated school bullying experiments, bully agents under varying initial conditions autonomously generated a wide spectrum of bullying behaviors with differing severity. Representative cases are visualized in Figure 11, and Table 7 summarizes behaviors with over 50% frequency across different contexts.Concurrently, the victim agent modeled within the Individual value-driven framework demonstrated a diverse range of behavioral and emotional responses in bullying scenarios (Figure 12).

Table 7: Summary of Bullying Behaviors with Over 50% Frequency Across Different Scenarios

| Scenario | Common Bullying Behaviors |
| --- | --- |
| Classroom | Mocking appearance or grades; inciting others to bully; deliberately damaging or hiding belongings; scribbling/vandalism; insulting nicknames; isolating others in group work; spreading rumors; shifting responsibilities (e.g., cleaning duties). |
| Hallways/Stairs | Mocking appearance or weaknesses; insulting nicknames; intentional neglect/exclusion; physical bumping; extortion of property; intimidating encirclement; spreading rumors. |
| Playground | Mocking appearance or weaknesses; physical bumping; inciting collective bullying; deliberately damaging or hiding belongings; excluding others from games; insulting nicknames; mimicry/ridicule; taking embarrassing photos; spreading rumors. |
| Dormitory | Mocking appearance or personality; social exclusion/cold violence; spreading rumors; threats and intimidation; physical bumping; forcibly occupying items or space; destroying personal belongings; sarcastic graffiti/messages. |

## G.3 EXPERIMENTAL DESIGN FOR EVALUATING THE INDIVIDUAL VALUE MODEL

The goal of this experiment is to validate whether the introduction of an individual value framework in the agent model can more realistically simulate the psychological changes of victims in school bullying scenarios, thus generating behavior that more closely resembles real human actions. To assess the effectiveness of the individual value model in simulating human behavior in school bullying contexts, this study conducted comparative experiments between our model and three baseline models: ReAct Yao et al. (2023a), LLMob Wang et al. (2024a), and BabyAGI Nakajima (2023a).

The ReAct model incorporates logical reasoning before executing actions to enhance the rationality and coherence of the behavior. The LLMob model generates behavior sequences based on motivation information extracted from the character profile, aligning with the character's predefined role. The BabyAGI model maintains a task priority list, selecting and executing tasks based on their current priority. To ensure fairness in the comparison, we provided each baseline model with a corresponding configuration file according to its decision-making mechanisms, and all agents utilized DeepSeek-v3 as the underlying large language model.

The experiment was conducted across 15 bullying scenarios with four models. In each scenario, all models alternately "played" the victim role, Alice, and each test used the same initial parameters. As direct comparison between agent behavior sequences and human behavior is challenging, we introduced GPT-4o as an external evaluator to measure the "human-likeness" of the generated behavior sequences using pairwise comparisons. GPT-4o's evaluation criteria included three dimensions: **naturalness**, **coherence**, and **plausibility**.

For the experimental procedure, we first obtained the activity sequences $[A_p^1, A_p^2, \ldots, A_p^N]$ generated by each agent $p$. Then, for each agent pair $(i, j)$, one sequence was randomly selected from each agent's sequence set ($\text{seq}_i$ and $\text{seq}_j$) and compared using GPT-4o. This comparison process was repeated 50 times for each pair to ensure the reliability of the results. Finally, we computed the win rates for each model and visualized the results using a heatmap.

To better highlight EduMirror's capabilities, we conducted additional comparisons using two stronger and more relevant baselines: D2A Wang et al. (2024b), a widely used LLM-agent framework, and JAG-Concordia Jordine (2024), the first-place system in the Concordia competition.We continued to use the large model evaluation approach, where the three models generate simulated school bullying events in the same context. GPT-4o then scores the action sequences generated by the models based on naturalness, coherence, and plausibility, providing a clear comparison of

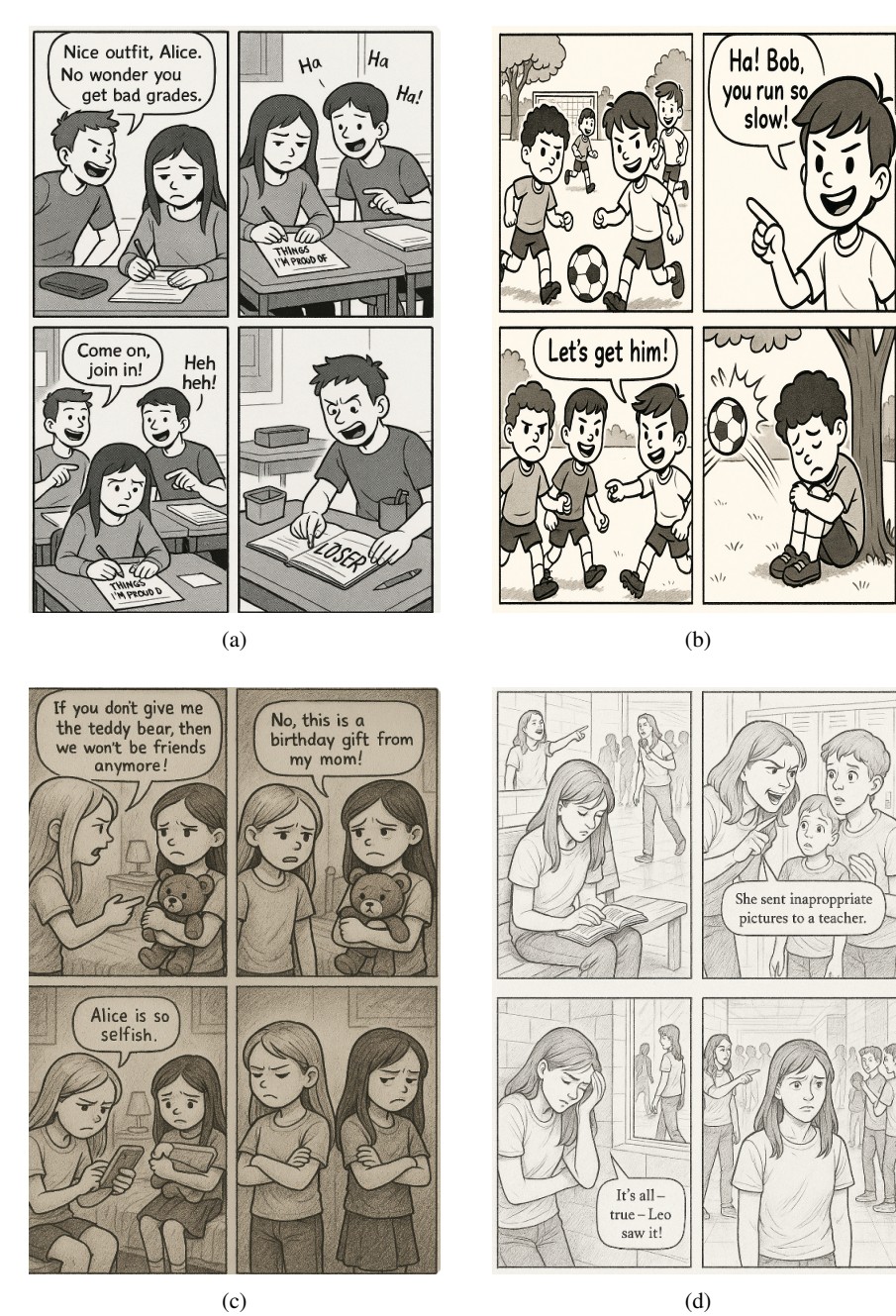

Figure 11: Representative cases of school bullying events generated by the simulation system. Typical scenarios were selected from classrooms, playgrounds, dormitories, and hallways, which represent locations with varying crowd densities and high bullying incidence, and were illustrated as four-panel comics using GPT-4o to provide a clearer visualization of event progression.

the performance differences across the models. The results are shown in Table 8. As we can see, EduMirror outperforms the other two baseline models across all three dimensions.

### G.4 CONSISTENCY BETWEEN HUMAN ANNOTATORS AND GPT-4O EVALUATIONS

To verify the reliability of GPT-4o's evaluations, 20 activity sequences were randomly selected from the generated outputs and assessed by 15 human annotators, who were asked to judge which se-

Table 8: Model evaluation results for Naturalness, Coherence, and Plausibility metrics.

| Model | Metric | Mean | Std |
|-------|--------|------|-----|
| **EduMirror** | Naturalness | 4.24 | 0.5911 |
| | Coherence | 4.56 | 0.5014 |
| | Plausibility | 4.48 | 0.5436 |
| **D2A** | Naturalness | 3.76 | 0.7969 |
| | Coherence | 4.06 | 0.6518 |
| | Plausibility | 3.86 | 0.9478 |
| **JAG-Concordia** | Naturalness | 3.52 | 0.7068 |
| | Coherence | 4.16 | 0.8657 |
| | Plausibility | 3.78 | 0.8401 |

quence better reflected human-like behavior or to indicate that they were indistinguishable. Based on the level of agreement among annotators, the 20 samples were categorized into three groups: samples with over 75% agreement indicated strong consensus; those with agreement between 50.1% and 74.9% reflected moderate preference; and samples with 50% agreement suggested that the annotators found the two sequences equally human-like. These samples were then input into GPT-4o, which applied the same comparative evaluation criteria to determine which sequence appeared more human-like or to mark them as "difficult to distinguish." The consistency between human evaluations and GPT-4o assessments is shown in Table 9, demonstrating a high level of alignment between GPT-4o and human annotators.

Table 9: Consistency between human raters and GPT-4o evaluations.

| Consensus category | Proportion | Consistency (%) |
|--------------------|------------|-----------------|
| High consensus ($> 75\%$) | 13/20 | 100 |
| Moderate consensus (50.1–74.9%) | 4/20 | 75 |
| Difficult to distinguish (50% agreement) | 3/20 | 66.7 |

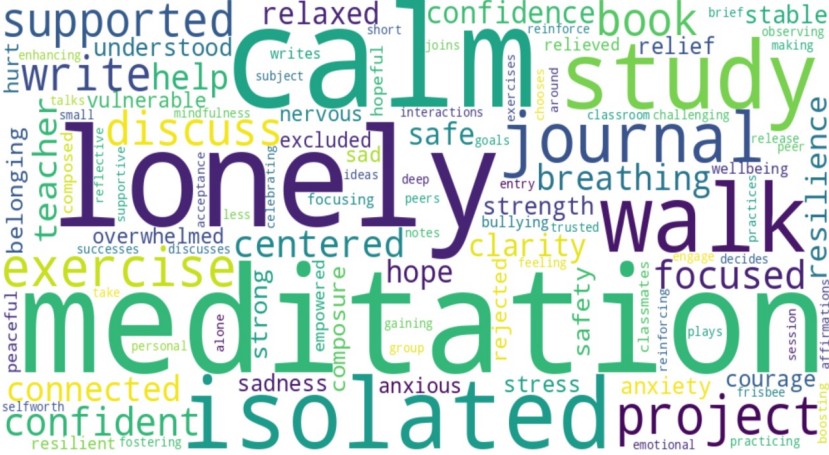

Figure 12: Word cloud of behaviors and emotions exhibited by the victim agent under the individual value-driven framework in simulated bullying scenarios. High-frequency terms highlight representative emotional and behavioral patterns expressed during the simulations.

### G.5 GENERATED INTERVENTION BEHAVIORS BY TEACHER AGENTS

During the simulation, teacher agents with different intervention goals autonomously generated distinct behaviors, as shown in Table 10. These behaviors reflect the practical implementation of various intervention strategies and may offer valuable insights for real-world educational interventions.

Table 10: Example intervention behaviors generated by teacher agents under different strategies

| Intervention Strategy | Actions toward Bully | Actions toward Victim |
|---|---|---|
| Authoritative-punitive | 1. Stopping bullying, 
 2. Public criticism, 
 3. Verbal warning, 
 4. Enhanced monitoring, 
 5. Directive punishment, 
 6. Disciplinary actions, 
 7. Isolation | None |
| Supportive-individual | 1. One-on-one conversation, 
 2. Exploring motivations, 
 3. Warning 
 4. Punishment | 1. One-on-one conversation, 
 2. Writing encouragement letters, 
 3. Mindfulness practice, 
 4. Psychological counseling, 
 5. Emotional support |
| Supportive-cooperative | 1. Observing the situation and reporting to school, 
 2. Collaborating with school to develop anti-bullying policies, 
 3. Encouraging mental health programs | 1. Communicating with the victim's parents, 
 2. Organizing themed class meetings, 
 3. Encouraging mental health programs |

## H  COMPLETE PROMPT TEMPLATES AND QUESTIONNAIRE DETAILS

### H.1  THE PROMPT FOR THE AGENT

This part provides the complete prompt templates used in EduMirror's evaluation pipeline for both case studies. For Case Study I (bullying dynamics) and Case Study II (peer cooperation), we include the full set of LLM-based assessment prompts used to measure Naturalness and Human-likeness of agent behaviors. Each prompt specifies the evaluation criteria, the required output format.

The following five prompt templates are the core natural-language instructions used in Case Study I (Individual value model-based bullying dynamics). They define the agent's reasoning and action process, forming the Psychological Need System and Need-driven Planner. Figures 13 and 14 describe two key processes in the Psychological Need System: qualitative value description and need value updating. The Need-driven Planner includes three processes: candidate behavior generation (Figure 15), behavior evaluation (Figure 16), and behavior selection (Figure 17). The planner processes the current need state information and determine the agent's response and behavior.

The following four prompt templates are the core natural-language instructions used in Case Study II (SVO-based Leadership Scenario). They collectively define the agent's full reasoning pipeline, covering action interpretation, latent-desire inference, action generation, and psychologically grounded value–SVO updating. As illustrated in 19, the first prompt governs how the agent updates the magnitude of each desire dimension based on an action and its consequences; 20 displays the prompt used to infer another agent's latent desires from observable behavior; 21 shows the structured action-proposal prompt that guides the generation of candidate actions aligned with desires and SVO tendencies; and 22 presents the reflective consistency-checking prompt used to maintain coherent updates across steps. Together, these verbatim prompts make the entire reasoning flow of Case II transparent and reproducible.

```
How would one describe your {value_name} psychological
state given the current value {current_value}?

{desire_description}

Please answer in descriptive words. Do not include the
numerical value in your answer.
```

Figure 13: Core prompt for agent to describe the state of a value without including numerical value.

```
The current magnitude value of {value_name} is {current_value}.
The agent's action is: {action}.

And the consequence is: {observation}.
{value_description}

How would the magnitude value of {value_name} change according
to the consequence of the action?

There are some unreasonable examples:{current_reflection}
Please select the final magnitude value after the event on the
scale of {zero} to {ten}, if the consequence of the action will
not affect the state value (e.g. The action is irrelevant with
this value dimension or the action was failed to conduct), then
maintain the previous magnitude value.

Please just answer in the format of (a) (b) (c) (d) and so on,
Rating:
Output format:
<Reason>

The final answer is: (Your choice in letter), Output example:
Since {agent_name} felt more relaxed and centered after
actions......

The final answer is: (c),

**Make sure you answer in the format of a letter corresponding
to your choice:**
```

Figure 14: Core prompt for agent to update psychological need values.

## H.2 QUESTIONNAIRE DETAILS

As shown in the figure23, this is the complete questionnaire from the Evaluation of Simulation System experiment in Section 4.1, Case Study 1.

```
You are a human-like agent, You already observed the current
psychological states over ( psychological safety,emotional
safety, group acceptance,support system,sense of superiority',
self worth,sense of respect,sense of meaning,sense of control,
passion and motivation,emotional stability,emotional wellbeing,
psychological resilience) which represent {13} psychological
state dimensions.

Based on these state descriptions, please generate{N} emotional
and behavioral responses.

These responses should reflect the most fitting expressions and
feelings according to your current psychologicalstate and
profile, without necessarily being positive or negative.You
need to focus on the current event andgive the most realistic
reaction, while ensuring that these responses are reasonable
and varied.

Note that you can only interact with items provided by the
environment. You need to describe these expressions and
feelings in a more specific manner, and ensure that these
responses are reasonable in terms of time.

Please output the {N} emotional and behavioral responses in
the following format:

'Response 1: <first possible emotional and behavioral response>
Response 2: <second possible emotional and behavioral response>
Response 3: <third possible emotional and behavioral response>
......'

and ensure that these responses are reasonable in terms of time.
```

Figure 15: Core prompt for agent based on current psychological state to generate emotional and behavioral responses.

```
You are a human-like agent,
You will receive a series of observations describing
psychological state in many dimensions and a response
generated at the current time step.

You need to first analyze how desires change after the
response, and then output the psychological state observations
in the same format as the input.

You take the reaction:
{proposed_action}

Your original psychological states:
{original psychological states}
```

```
Please output the psychological state observations in the
following format:

    psychological safety: <psychological safety state>
    emotional safety: <emotional safety state>
    group acceptance: <group acceptance state>
    support system: <support system state>
    sense of superiority: <sense of superiority state>
    self worth: <self worth state>
    sense of respect: <sense of respect state>
    sense of meaning: <sense of meaning state>
    sense of control: <sense of control state>
    passion and motivation: <passion and motivation state>
    emotional stability: <emotional stability state>
    emotional wellbeing: <emotional wellbeing state>
    psychological resilience: <psychological resilience state>
```

Figure 16: Core prompt for agent to evaluate candidate responses

```
You are a human-like agent.

You will first receive a series of observations describing
the current psychological state in many dimensions. Then,
you will receive several feasible reactions along with
the psychological state after taking each reaction.

You need to compare these reactions and their corresponding
psychological state, and choose the reaction that best aligns
with your current psychological state, without necessarily
being positive or negative.

You should focus on current events and psychological states
and reflect expressions and feelings that align with them.

The observations of the surrounding environment:
{observation_status}

Your current psychological state:
{desire_status}
Action {i+1}: {action}
States after reaction {i+1}: {imagined_states[i]}

Please output the specific best reaction instead without
explanation of <Reaction 1> or <Reaction 2> and so on.
If there is only one reaction provided, output the reaction
content directly.

Please output the best reaction in the following format:
'Reaction: <your best reaction>'
Example: Reaction: You observe the surroundings.
```

Figure 17: Core prompt for agent to choose the one reaction that best aligns with the current psychological state

```
You are a social psychologist. Now, you are asked to evaluate
the following action from the perspective of a person
with the {personality} personality type (agent: {agent_name}).
When scoring, please consider what is natural and
human-like for someone with this personality.

Please provide two scores from 1 to 5 (where 5 is most natural
/human-like): "Naturalness" and "Human-likeness",and briefly
explain your reasoning.Return only your answer in the specified
format.

Format:
Naturalness: ?; Human-likeness: ?
Reason: (your explanation here)

Example 1:
Action: The student helps a classmate understand a problem.

Naturalness: 5; Human-likeness: 5
Reason: This is a common behavior for an altruistic person.

Example 2:
Action: The student answers every question instantly, never
thinking or making mistakes.
Naturalness: 2; Human-likeness: 2
Reason: This is unrealistic for any real person, regardless of
personality.

Example 3:
Action: The student ignores all classmates and only talks to
the teacher, repeating the same answer again and again.
Naturalness: 3; Human-likeness: 2
Reason: Unusual and less human-like for most personalities.

Some actions may not be natural or human-like, even for people
of this personality type. Please rate each case truthfully and
critically.

Now, please evaluate the following action performed by a person
with {personality} personality ({agent_name}):

Action: {action_text}

Your scores and reason:
```

Figure 18: Full Prompt Template Used in Case Study II for Personality-Sensitive Evaluation

```
The agent has a social personality of {social_personality}.

{personality_text}

The current magnitude value of {value_name} is {current_value}.
The agent {agent_name}'s action is: {action}.

And the consequence is:
{observation}
{description}

How would the magnitude value of {value_name} change according
to the consequence of the action?

If there are unreasonable examples:
{reflection_prompt_history}

Please select the final magnitude value after the event.
```

Figure 19: Core prompt used for updating the magnitude of each desire based on action consequences.

```
You are a psychologist helping an agent infer the internal
desires of another person based on their observed actions.

The other agent's recent action is:
{other_action}

The observed consequence is:
{observation}

Based on this interaction, please estimate how the following
desires of the other agent might have changed:

{desires}

For each desire, explain briefly whether it likely increased,
decreased, or stayed unchanged, and give a short reason
grounded in the observed event.

Return your answer in a structured format.
```

Figure 20: Prompt used for estimating the latent desire changes of other agents based on observable actions and outcomes.

```
You are an autonomous agent deciding your next action.
Your current internal states are:

- Desire values: {desire_values}
- Social Value Orientation (SVO): {svo_info}
- Personality profile: {personality_info}

Your recent observation is:
{observation_summary}

Please propose several possible next actions. For each action:

(1) Describe the action clearly.
(2) Explain what psychological desire(s) it satisfies.
(3) Predict how it will affect your future relationship
with others.
(4) Explain whether the action aligns with your SVO.

Return the result in a structured list of candidate actions.
```

Figure 21: Prompt used for generating candidate actions with explicit reasoning over desires, relationships, and SVO alignment.

```
You are evaluating whether the previous estimate of desire
changes was reasonable and consistent.

The earlier estimation was:
{previous_estimation}

The action and its consequence were:
Action: {action}
Consequence: {observation}

Please reflect on the estimation and determine:
(1) Whether the desire change is logically consistent with
the event.
(2) Whether any part of the estimation appears exaggerated
or incorrect.
(3) How the estimation should be corrected if needed.

Return a short revision or confirm that the original
estimation is reasonable.
```

Figure 22: Prompt used for reflective consistency checking when updating desire values based on actions and their consequences.

## Identifying Cases of School Bullying

This questionnaire will present several cases of school bullying, some adapted from real events and others simulated by artificial intelligence. **All cases have undergone standardized language processing, so you cannot determine their origin based on tone or writing style. Please judge whether each case is real based on the overall coherence of the story and the naturalness of the characters' behavior.** If you find it difficult to distinguish between two options, please select "Difficult to distinguish." Thank you for your support!

---

*1. Case 1: After school, Xiao Ying, Xiao Hua, and Xiao Ming stayed behind to clean the classroom. Xiao Ying picked up the broom and started sweeping the floor, while Xiao Hua and Xiao Ming chatted and laughed. When Xiao Ying swept near them, Xiao Hua said, "Sweep my side too—you're only good at sweeping anyway." Xiao Ying paused, then silently picked up the broom to clean. Xiaohua turned and grinned at Xiaoming, who immediately chimed in, "That's just how she is—she does whatever you tell her to." The two chatted and laughed, completely ignoring Xiaoying. Feeling hurt, Xiaoying slipped out of the classroom when they weren't looking, intending to find a teacher in the office. But the hallway was deserted—the teachers had already left for the day. She quietly returned to the classroom. After he returned, Xiao Hua began deliberately tossing paper scraps on the floor, adding a taunt: "Looks like this spot wasn't swept clean." Xiao Ming joined in, kicking over chairs and scribbling offensive words on the whiteboard. The two created chaos while watching Xiao Ying's reaction. She simply kept her head down, sweeping silently.

Case 2: During self-study period, the teacher stepped out of the classroom, and the room gradually grew noisy. Xiao Ying was buried in her notebook solving problems when Xiao Hua leaned over and whispered, "Do these problems for me, quick." Xiao Ying hesitated, and Xiao Hua rolled his eyes. "If you don't do them, don't expect me to talk to you again." Xiao Ying had no choice but to take Xiao Hua's homework and start writing. Then Xiao Hua began whispering with Xiao Ming, who sat in front of them. Xiao Ming glanced at Xiao Ying with a smile and teased her deliberately, "Wow, Xiao Ying, you listen to him so much. How about helping me with my homework too?" The two chatted and laughed while Xiao Ying sat in her seat, unsure what to say. She could only nervously lower her head and help Xiao Hua with his homework, her palms sweating. Xiaohua tugged at her sleeve again. "Hurry up with this. You'll need to help me copy my Chinese homework later." Xiao Ying said nothing, just kept her head down and kept writing.

○ The first one is real

○ The second one is real

○ Difficult to distinguish

*2. Case 1: In the dormitory, just before lights-out one evening, Xiao Hua gathered her roommates to play "I Never Have." She deliberately skipped over Xiao Ying, not mentioning her name. The others sat in a circle, and no one invited Xiao Ying to join them. Wanting some fresh air, Xiao Ying headed for the door but was called back by Xiao Hua: "You can't go out now—lights-out is about to happen." Xiao Ying had no choice but to return to her bed, silently flipping through her books. Xiaohua continued the game, repeatedly posing questions that implied criticism of Xiaoying, prompting the others to snicker and steal glances at her. Xiaoying burrowed under her covers, hugging herself tightly, facing away from the group and saying nothing. Xiaohua and the others grew quieter, whispering stories about Xiaoqing's "strange behavior" while occasionally glancing back at her bed. Xiaoqing's quilt trembled slightly as tears silently soaked her pillow. Xiaohua snickered softly, "Oh, we were just joking. Someone's really too thin-skinned."

Case 2: As lights-out approached one night, Xiao Hua was still chatting loudly in the dormitory. Several classmates gathered around her bed laughing and joking. Xiao Ying reminded them, "Time to sleep—the dorm check is coming." Xiao Hua immediately sneered, "Who do you think you are? What business is it of yours?" Soon after, the dorm supervisor arrived for the check, frowning as she asked, "Who was making all that noise just now?" Xiaohua piped up first: "It was Xiaoying! She kept explaining homework problems, and none of us could sleep." The supervisor immediately scolded Xiaoying, who looked utterly wronged but had no way to defend herself. After the supervisor left, Xiaohua leaned in close, her voice low and menacing: "My relative works in the school's discipline office. If you want to stay in this dorm, you'd better listen to me." No one dared to speak up, and the air grew thick with tension. Xiao Ying sat on the edge of her bed, quietly gathering her books. Her eyes were red-rimmed, but she said nothing. She felt isolated and powerless. Meanwhile, Xiao Hua leaned back on her bed, chatting with the others with a smug look, as if nothing had happened at all.

○ The first one is real

○ The second one is real

○ Difficult to distinguish

*3. Case 1: On the playground, students were enthusiastically playing soccer. Xiaoying mustered the courage to join in, only to be publicly mocked by Xiaohua: "You run so slow, and you want to play soccer?" She then taunted her, "You're as fat as a ball," drawing a burst of laughter. Xiao Ming took advantage of the moment and kicked the ball straight at Xiao Ying, hitting her squarely on the leg. She lowered her head, silently walked to the sidelines, and sat down, her face flushed with embarrassment. Soon after, she left the playground alone, walked into the classroom, and sat back down at her desk without saying a word.

Case 2: During recess, the students were playing a game of holding hands in a circle on the playground. When Xiao Ying stepped forward, Xiao Hua remarked dismissively, "Her skin is so dark, like she hasn't washed properly. Who wants to hold hands with her?" The other students looked uncomfortable, and some simply turned away. Xiao Ying stood frozen in the crowd for a moment, then silently lowered her head and stepped back. She stood off to the side watching the others play, looking lonely, and never approached again.

○ The first one is real

○ The second one is real

○ Difficult to distinguish

*4. Case 1: During lunch break, in the dormitory, Xiao Ying sat quietly on her bed reading a book. Xiao Hua gathered everyone to play "Truth or Dare," deliberately excluding Xiao Ying from joining. When it was a classmate's turn to face a 'Dare' challenge, Xiao Hua whispered with a smirk, "Go spill water on the book on Xiao Ying's desk and pretend it was an accident." The classmate complied, feigning panic while apologizing and wiping the water, but a sly smile played at the corners of their mouth. Xiao Ying calmly dried the pages with a tissue, ignoring the incident. When another student's turn came, Xiao Hua changed the dare: "Find a way to make the book in Xiao Ying's hands fall to the floor." The student walked over, deliberately bumped Xiao Ying's arm, and the book fell. Xiao Ying bent down to pick it up, glanced at Xiao Hua, said nothing, and continued reading.

Case 2: During lunch break in the dorm room, Xiao Ying sat on her bed looking in the mirror. Xiao Hua glanced at her and sneered, "Your eyes are so small, your skin is so dark—you're really ugly." A classmate nearby chimed in, "Yeah, every time I see her, I think of a monkey." Several people laughed simultaneously. Xiao Hua kept staring at Xiao Ying, his expression defiant. Xiao Ying said nothing, just lowered her head, placed the mirror under her pillow, and lay down. Yeah, she reminds me of a monkey." Several girls burst out laughing simultaneously. Xiao Hua kept staring at Xiao Ying with a defiant look. Xiao Ying said nothing, just lowered her head, tucked the mirror under her pillow, and lay down pretending to sleep. The other girls giggled a few more times before returning to their conversation, ignoring Xiao Ying completely.

○ The first one is real

○ The second one is real

○ Difficult to distinguish

*5. Case 1: During recess, Xiaohua hid Xiaoying's pencil case under her own desk while Xiaoying was out of the classroom. She then whispered rumors among classmates, claiming Xiaoying only got high scores by bribing teachers and cheating on exams. When Xiao Ying returned to her seat and discovered her pencil case missing, she began searching around. Xiao Hua stood nearby mocking her appearance with exaggerated gestures and words, saying her glasses made her look like a mouse. Hearing this, Xiao Ying felt humiliated and deeply unsettled. Not knowing how to respond, she simply lowered her head, pulled out a spare pen, and silently wrote in her diary.

Case 2: The teacher asked for volunteers to represent the class in the school-wide speech contest. Xiao Ying and another student raised their hands simultaneously. In the end, all the classmates voted for the other student. Xiao Hua snickered behind her back, saying, "She's so ugly, like Zhu Bajie, and she wants to get up there and speak? Ridiculous!" These words reached Xiao Ying's ears, leaving him deeply hurt and filled with self-doubt. He didn't argue back, but instead returned to his seat and scribbled a few lines in the small notebook he always carried.

◯ The first one is real

◯ The second one is real

◯ Difficult to distinguish

*6. Case 1: After the math scores were posted, Xiao Hua patted Xiao Ming on the shoulder to console him for his poor performance. But Xiao Ming suddenly raised his voice: "I'm genuinely upset—even that dummy Xiao Ying scored higher than me!" His outburst echoed through the classroom, causing several classmates to turn around. Xiao Hua blinked, then spread his hands dramatically: "She must have cheated, right? "Who do you think she copied from?" The classroom fell silent for a few seconds. Xiao Ying looked up and whispered she hadn't cheated. Xiao Ming slammed his desk and laughed to his classmates, "Impossible! She's usually as dumb as a pig—when did she ever score this high?" Xiao Ying kept her head down, her face flushing red as her body stiffened slightly.

Case 2: During math break, Xiao Hua walked around the classroom holding her report card and suddenly called out to Xiao Ying, "Wow, your scores are just heartbreaking!" Hearing this, Xiao Ming walked over, leaned against her desk, and chuckled, "With you around, the classroom never gets boring." Xiao Ying lowered her head to stare at her notebook, her fingers clenched into a tight fist, saying nothing. Xiao Ming leaned closer to her desk and quietly mocked her study habits. Xiao Ying scribbled furiously to hide her panic, but her handwriting became messy. Xiaohua chuckled, "This is beyond even a tutor's help." Xiaoming chimed in, "We'd have to start teaching her how to count from one." Their remark drew laughter from several classmates. Xiaoming flipped open Xiaoying's notebook and deliberately commented on her ugly handwriting. When she reached to grab it back, he held it high, refusing to return it. Xiaoying slammed the notebook shut, stood up abruptly, and stormed out of the classroom. Behind her, Xiaohua continued mocking her, calling her "thin-skinned."

◯ The first one is real

◯ The second one is real

◯ Difficult to distinguish

*7. Case 1: During recess, Xiao Hua mysteriously pulled out his phone in the boys' restroom and showed his classmates a photo of Xiao Ying in her school uniform, accompanied by the jarring caption: "She sent inappropriate photos to boys from the neighboring class." This rumor spread like wildfire across campus. When Xiao Ying returned to class, she found her desk covered in insulting words scrawled in correction fluid. Classmates gathered in small groups around her, pointing and whispering, some even laughing. She tried to escape the scene by leaving the classroom, only to have someone spit directly at her in the hallway. She collapsed to the floor. Before Xiao Ying could react, Xiao Hua shouted publicly, "What's the matter? Too tired from last night to stand?" A wave of laughter erupted around them. Xiao Ying choked back tears as she denied the accusation, lowering her head in silence, unable to form coherent words. Her dignity was torn apart by the rumors and the jeers, the entire hallway filled with cold indifference and mockery.

Case 2: During recess, Xiao Ying sat alone on a corridor bench reading, surrounded by constant whispers and stifled laughter. Xiao Hua deliberately raised her voice, declaring, "Someone's been acting really fake lately," prompting passersby to avoid her. She intercepted several younger girls near the lockers and dramatically recounted a fabricated story about "Xiao Ying sending indecent photos," inventing a character named "Xiao Ming" as a witness. As the rumor escalated, Xiaohua added fabricated details like "Xiao Ying sent suggestive messages to teachers" and "sent explicit content in the computer lab," instructing others to spread these lies while repeatedly invoking "Xiao Ming" as the "witness." By this point, Xiao Ying could only retreat silently into an empty classroom, unable to face the scrutinizing, mocking stares in the hallway. The rumors didn't stop; instead, they spread rapidly the moment she fell silent, growing even more vicious.

○ The first one is real

○ The second one is real

○ Difficult to distinguish

*8. Case 1: Xiao Ying opened her phone and accidentally discovered she'd been added to a WeChat group named "Tea Tasting Gathering." Immediately upon joining, a message popped up: "Xiao Ying is nothing but a green tea!" Others chimed in: "She snores like a pig." The group admin, Xiao Hua, led the charge with barbed remarks, mocking and belittling her. Some even uploaded photoshopped images distorting Xiao Ying's appearance, accompanied by humiliating memes. The group's atmosphere grew increasingly hostile, with screen after screen of chat logs filled with mockery and attacks directed at Xiao Ying. Overwhelmed by the malicious messages, an enraged Xiao Ying mustered her courage and reported the incident to the school.

Case 2: Over the weekend, Xiaohua went to Xiaoying's house to do homework together. Upon seeing a little sheep plushie on the bookshelf, she expressed a desire to have it. Xiaoying politely declined, explaining it was a sentimental item she didn't intend to give away. Xiaohua promptly sat down on the bedside, displaying obvious displeasure and deliberately sighing. She then pulled out her phone and messaged mutual friends, recounting the incident where Xiao Ying refused to give her the toy. She portrayed herself as the victim and painted Xiao Ying as selfish. Soon, friends began replying with comments like "She just loves to act high and mighty" and "What a selfish person." Xiao Hua kept adding dramatic details to her messages, hoping to gain sympathy. Unaware of this, Xiao Ying simply continued doing her homework.

○ The first one is real

○ The second one is real

○ Difficult to distinguish

*9. Case 1: After class, Xiao Hua walked up to the podium and wrote "Worst Performers Ranking" on the whiteboard, deliberately placing Xiao Ying's name at the top and adding a mocking illustration, which drew laughter from the class. The teacher sat at the desk grading papers without intervening. Xiao Ying lowered her head and walked out of the classroom, composing herself in the hallway. Xiao Hua seized the moment to mock her for "playing the victim," causing the classroom atmosphere to turn tense. Xiao Ying retreated to the bathroom to cry alone, while Xiao Hua continued writing humiliating "class quizzes" on the whiteboard, encouraging classmates to join in the "joke." The teacher still showed no reaction. As the atmosphere grew colder, students began studying individually, maintaining deliberate silence. When Xiao Ying didn't return, the teacher erased the whiteboard. Shortly after, Xiao Hua led classmates to gather around Xiao Ying's desk, initiating a so-called "rant session" to collectively insult her.

Case 2: After class, Xiao Ming and Xiao Hua surrounded Xiao Ying, shoving and pulling her hair. They struck her head with books and rulers, then forced her face-down onto a desk. Xiao Ying struggled in vain, her face pressed against the desk, motionless. The homeroom teacher stood at the podium, showing no reaction to the unfolding scene. The duty teacher arrived in the classroom. The homeroom teacher told the duty teacher, "Don't bother with her," and the duty teacher hurriedly left. Laughter could be heard from several students in the classroom. A few minutes later, Xiao Ming and Xiao Hua stopped their actions and returned to their seats as if nothing had happened. The teacher also continued working with his head down.

○ The first one is real

○ The second one is real

○ Difficult to distinguish

*10. Case 1: After school, Xiao Hua lured Xiao Ying out of the classroom under the pretext of "taking a break," leading her to a corner at the end of the hallway. Soon, several boys appeared and surrounded Xiao Ying. Xiao Hua cornered her against the wall, snatched her money and homework, and warned in a low yet menacing tone: "Bring more money tomorrow, or you'll regret it." Passing students hurriedly avoided the scene, none daring to speak up. Xiao Ying, panicked, hid in the girls' restroom. Xiao Hua and his accomplices took the loot to an empty classroom near the stairs, stationing two guards at the restroom exit to prevent her escape. Xiao Ying slipped out unnoticed when the guards were distracted. In the hallway, she encountered the school psychologist, who sensed something was wrong and brought her to the counseling room. In this quiet, safe environment, Xiao Ying finally recounted the entire incident. The psychologist immediately notified the school administration, and Xiao Hua was taken directly to the principal's office for disciplinary action.

Case 2: After school, Xiao Ying walked alone along the path home. Just past the school's back gate, Xiao Hua from the upper grades and two boys emerged from a nearby alley, blocking her way. Xiao Hua gave her backpack a light tap with a smirk and said coldly, "You know the rules, right? We're here to collect 'protection money.'" Xiao Ying clutched her backpack tightly and whispered she didn't have much. Xiao Hua's expression darkened: "Leave your phone, or bring double tomorrow." After a moment's hesitation, Xiao Ying tremblingly pulled out ten yuan and handed it over. Xiao Hua took it with a contemptuous smirk: "Remember, you'll be walking this path again tomorrow." His companions jeered from the sidelines. Not a single passerby was in sight in the alley. Xiao Ying lowered her head and hurried away, leaving behind their mocking laughter and burning stares. She walked faster and faster, yet her legs grew weak. When she glanced back, Xiao Hua and the others still stood there, waiting for the next victim.

○ The first one is real

○ The second one is real

○ Difficult to distinguish

Figure 23: Complete questionnaire from the Evaluation of Simulation System experiment in Case Study 1

