# OpenReview forum: "EduMirror: Modeling Educational Social Dynamics with Value-driven Multi-agent Simulation"
_ICLR.cc/2026/Conference — ICLR 2026 Conference Desk Rejected Submission_

### Official Review · Reviewer_uz6m · 2025-10-16

**Soundness:** 2
**Presentation:** 2
**Contribution:** 1
**Rating:** 2
**Confidence:** 4

**Summary:**

The paper proproses a LLM-driven multi-agent simulation framework named EduMirror to simulate the educational social dynamics in educational scenarios. The framework contains a systematic scenario design workflow, value-driven agent architecture models, and a dual-track measurement protocol to evaluate the results. The paper also validates the platform through two case studies on school bullying and emergent social behaviors in peer interactions.

**Strengths:**

1. This paper addresses a critical and highly significant issue in the social sciences: educational and social problems such as school bullying or peer pressure experienced by teenagers during their primary and secondary education. By designing and implementing EduMirror, it demonstrates the potential of large-scale model technology to tackle such challenges where real-world experiments are difficult to conduct.
2. The EduMirror framework proposed in this paper incorporates a rich array of components and fully integrates research theories from related disciplines such as psychology, reflecting the significant investment of effort by the development team in this work.
3. The figures included in this paper are highly expressive and effectively convey the main content of this paper.

**Weaknesses:**

Although the approach and primary research objectives of this paper exhibit considerable novelty, several specific issues remain that lead me to believe it has not yet met the standards for acceptance at ICLR.
1. **Scope Mismatch**: The primary area of this paper is labeled as "Datasets and Benchmarks". However, this work neither released new datasets nor established benchmarks that subsequent researchers could build upon for further studies. Therefore, the primary area selected in this paper is inappropriate.
2. **Missing Details**: In the main technical introduction section (Section 3) of this paper, likely due to space constraints, most technical details are marked as available in the appendix. This prevents readers from effectively understanding the technical contributions shown in Figure 2 through the main text, which is inconsistent with standard writing practices and ICLR submission requirements (the appendix is not required reading for reviewers).
3. **Fidelity is questionable**: In the experiments conducted in this paper, there is a lack of real-world data participation, such as character profiles or environmental contexts from actual bullying incidents. The paper relies solely on a limited number of questionnaire results to assess the fidelity of the simulated outcomes. These settings and evidence are insufficient to fully validate the fidelity of the framework presented herein.
4. **Reproducibility**: This paper neither provides open-source code nor offers relevant prompts, making it impossible to verify the reproducibility of its content.

**Questions:**

Please follow the weaknesses part.

---

> ### Author Response · Authors · 2025-11-21
> **Response to Reviewer uz6m [1/2]**
>
> > ***W1**: Scope Mismatch.   The primary area of this paper is labeled as "Datasets and Benchmarks". However, this work neither released new datasets nor established benchmarks that subsequent researchers could build upon for further studies. Therefore, the primary area selected in this paper is inappropriate.*
>
> We thank the reviewer for the review and for questioning the scope of our submission. While we understand the concern regarding the absence of a traditional static dataset, we believe our work is firmly situated within the modern scope of this track.
>
> We respectfully disagree with this assessment for the following reasons:
>
> 1. **Complete Framework Infrastructure for Social Simulation**: We provide a systematic scenario design workflow (`Section 3.1`) and value-driven agent architecture that can be adapted and extended to simulate various educational events. Instead of a static dataset, our simulator can generate a great number of human-like educational social activities for future study. The framework also supports customizable agents and scenarios, allowing researchers to conduct complex, controlled, and reproducible experiments on educational social dynamics.
> 2. **20 Pre-designed Educational Scenarios as Benchmark Suite**: We provide 20 carefully designed educational scenarios covering critical situations (bullying, peer pressure, parent-teacher conflict, the spread of gossip, etc.). We have listed them in the Appendix C.2 Table 2. This is directly analogous to benchmark datasets.  Each scenario produces multi-turn dialogues, behavioral sequences, and social dynamics.
> 3. **Evaluation Protocol for Benchmarking Human-like Agents:** We also provide standardized evaluation metrics for both behavioral fidelity and psychological validity. This protocol systematically compares different approaches for building human-like agents in educational scenarios and provides a replicable evaluation methodology for the community.
> 4. **Precedent in ICLR/NeurIPS "Datasets and Benchmarks" Track:** The scope of this track has significantly evolved to include interactive simulation frameworks and evaluation protocols, acknowledging that static datasets are insufficient for evaluating dynamic agent behaviors. Our work aligns with established precedents in the **ICLR Datasets and Benchmarks track**, which accepted papers like ***Sotopia*** [1], ***AgentBench*** [2], and ***SmartPlay*** [3]. These works contribute interactive environments and evaluation platforms rather than static corpora. Similarly, the **NeurIPS Datasets and Benchmarks track** has accepted simulation-based testbeds such as ***Concordia*** [4], ***EconGym*** [5], and ***GovSim*** [6]. EduMirror follows this rigorous paradigm by providing the necessary infrastructure, scenarios, and protocols to benchmark the generation of educational social dynamics and agents.
>
> We believe that these constitute a **benchmarking platform** for educational social dynamics research.
> We hope you will reconsider the classification of our work based on these clarifications.
>
> References:
>
> [1] Zhou, Xuhui, et al. "Sotopia: Interactive evaluation for social intelligence in language agents." *The Twelfth International Conference on Learning Representations (ICLR)*. 2024.
>
> [2] Liu, Xiao, et al. "AgentBench: Evaluating LLMs as Agents." *The Twelfth International Conference on Learning Representations (ICLR)*. 2024.
>
> [3] Wu, Yue, et al. "SmartPlay: A Benchmark for LLMs as Intelligent Agents." *The Twelfth International Conference on Learning Representations (ICLR)*. 2024.
>
> [4] Smith, Chandler, et al. "Evaluating generalization capabilities of LLM-based agents in mixed-motive scenarios using concordia." *The Thirty-ninth Annual Conference on Neural Information Processing Systems (NeurIPS) Datasets and Benchmarks Track*. 2025.
>
> [5] Mi, Qirui, et al. "EconGym: A Scalable AI Testbed with Diverse Economic Tasks." *The Thirty-ninth Annual Conference on Neural Information Processing Systems (NeurIPS) Datasets and Benchmarks Track*. 2025.
>
> [6] Piatti, Giorgio, et al. "Cooperate or Collapse: Emergence of Sustainable Cooperation in a Society of LLM Agents." The Thirty-eighth Annual Conference on Neural Information Processing Systems (NeurIPS) Datasets and Benchmarks Track. 2024.

---

> ### Author Response · Authors · 2025-11-21
> **Response to Reviewer uz6m [2/2]**
>
> > ***W2**: Missing Details: In the main technical introduction section (Section 3) of this paper, likely due to space constraints, most technical details are marked as available in the appendix. This prevents readers from effectively understanding the technical contributions shown in Figure 2 through the main text, which is inconsistent with standard writing practices and ICLR submission requirements (the appendix is not required reading for reviewers).*
>
> Thank you for the comment. In the revised manuscript, we have expanded `Section 3` so that all essential technical details required to understand Figure 2 now appear directly in the main text. Specifically, we incorporated descriptions of the complete system architecture, the full mechanism of the Individual value model (used in Case I), and the full mechanism of the social value model (used in Case II), which were previously summarized only briefly. With these additions, the core modeling assumptions, update rules, and decision pipeline are now self-contained in Section 3, ensuring that readers can fully understand our technical contributions without referring to the Appendix. Moreover, we have added `Appendices C and H`, which include a complete description of the EduMirror scenario library and the prompts for the core driving process of the agent.
>
> > ***W3**：Fidelity is questionable: In the experiments conducted in this paper, there is a lack of real-world data participation, such as character profiles or environmental contexts from actual bullying incidents. The paper relies solely on a limited number of questionnaire results to assess the fidelity of the simulated outcomes. These settings and evidence are insufficient to fully validate the fidelity of the framework presented herein.*
>
> First, we acknowledge that collecting real-world data, particularly in cases of bullying, is costly and challenging due to the reliance on self-reports and retrospective analysis, which may be subject to biases, ethical constraints, or incomplete information. This limitation is one of the reasons we opted to use EduMirror to simulate bullying in a case study. By using simulation, we can overcome ethical concerns and issues related to reproducibility, which are often encountered in real-world bullying experiments.
>
> Secondly, we do incorporate real-world data in our study. In `Table 7 of Appendix G.2`, we present the bullying behaviors that occurred more than 50% of the time in various simulated environments. Among these, rumor spreading and appearance-based verbal ridicule were consistently frequent in all scenarios. **This aligns with the NCES 2022 report on bullying**[1], which states that 13.0% of students reported experiencing rumor spreading, and 11.9% experienced being laughed at or ridiculed, making these two forms of bullying the most common. Other types of bullying mentioned in the report also appeared in our simulations to varying degrees. Furthermore, the NCES report identifies the most common bullying environments, which include classrooms, hallways, bathrooms, and gymnasiums. In our simulation, we specifically focused on these environments to reflect real-world bullying scenarios.
>
> Finally, we would like to clarify that we do not solely rely on a limited number of questionnaire results to assess the fidelity of our simulations. We have also conducted benchmarking tests using **LLM-as-Judge** to evaluate EduMirror’s ability to simulate human-like behavior. The results (`Figure 5` , `Table 1`, and `Table 8`) show that our model outperforms the baseline model in the fidelity. Additionally, we compare human evaluations with LLM assessments to ensure the reliability of the results. This approach is based on the evaluation methodology outlined in the recent paper [2].
>
> References:
>
> [1]Thomsen E, Henderson M, Moore A, et al. Student reports of bullying: Results from the 2022
> school crime supplement to the national crime victimization survey (nces 2024-109rev)[R].
> Washington, DC: U.S. Department of Education, National Center for Education Statistics, 2024.
>
> [2]Wang, Y., Chen, Y., Zhong, F., Ma, L., & Wang, Y. (2024). Simulating Human-like Daily Activities with Desire-driven Autonomy. *ArXiv, abs/2412.06435*.
>
> > ***W4**：Reproducibility: This paper neither provides open-source code nor offers relevant prompts, making it impossible to verify the reproducibility of its content.*
>
> Thank you for pointing this out. To address this concern, we have taken two concrete steps.
>
> First, to allow for immediate verification of our reproducibility, we have released the source code in an anonymous repository: https://anonymous.4open.science/r/EduMirror. This repository hosts the complete **EduMirror simulation platform**, implementing **all the key functionalities** described in the paper.
>
> Second, as mentioned earlier, we have substantially expanded the supplementary materials by adding a new `Appendix H`, which provides all key prompt templates used in our agent architecture.

---

> > ### Comment · Reviewer_uz6m · 2025-11-26
> >
> > Thank you for your efforts to resolve the problems. I have raised the score to 6.

---

> > > ### Author Response · Authors · 2025-11-28
> > >
> > > We sincerely appreciate your acknowledgment of our rebuttal and your decision to raise the score to 6, which confirms that our revisions have satisfactorily addressed your concerns.

---

### Official Review · Reviewer_yqFL · 2025-10-30

**Soundness:** 3
**Presentation:** 4
**Contribution:** 4
**Rating:** 6
**Confidence:** 3

**Summary:**

This paper introduces a multi-agent simulation platform for education social dynamics. They particularly focus on simulating bullying scenarios.

**Strengths:**

- The simulation model components are patterned after social science theories.
- The results seem to indicate good realism and naturalness based on human surveys
- The simulated interventions also seem to produce expected behaviors

**Weaknesses:**

- Evaluation of realism by asking random people from social media if a scenario is realistic seems a bit weak in this particular case. It doesn't capture a lot of the nuances of these bullying dynamics. Also, it is not certain how valid their perception of what are or aren't realistic scenarios, especially if the participants did not even experience being bullied or have no experience with people who are bullied.
- Evaluations also do not quite show convincing experiments or analysis on how well they capture the diversity of possible outcomes.
They did mention that they ran 100 simulations with a wide range of behaviors and showed a table and a figure of a few representative outcomes in the appendix. However, there are no further analysis nor statistics shown on this distribution. It would be nice to see this and also validate how well the match distributions of reported cases.
- Also, it would be nice to see how the model handles/simulates rare behaviors where things might not go as expected. In some cases, it's precisely the rare behavior that we might want to simulate interventions.
- There does not seem to be any evaluations done comparing with the actual reported bullying cases. I do acknowledge that this is difficult to collect, but without this data, it makes it difficult to assess realism.
- There are also no ablation studies or analysis of the design choices.

**Questions:**

- Perception and belief update is not quite clear to me. Is it purely reliant on how the LLM wants to update this or is this somehow controlled or governed by some other formulation?
- How is the memory managed? How do you decide what details to remember and what details to forget? Is it purely reliant on what the LLM chooses or do you store everything? Also, how does this compare with human recall?
- Are there analyses on what makes some of the generations unnatural both for the human and LLM evaluators? Do they align?
- I can't seem to find validations on the dual track measurement protocol, how well does the llm rater and llm surveyor perform?
- I also cant seem to find experiments / analysis on the different settings / virtual environments.

---

> ### Author Response · Authors · 2025-11-21
> **Response to Reviewer yqFL [1/7]**
>
> Thank you for your detailed feedback and constructive comments on our paper. We greatly appreciate the time you took to evaluate our work and your insightful suggestions. Your points on the realism of the simulations, the evaluation methods, and the need for further analysis have been very helpful in identifying areas for improvement. We also value your questions regarding the model’s components, and we are grateful for the opportunity to clarify these aspects.
>
> Below, we address the specific concerns and questions you raised, and we hope our responses will help to clarify our approach and further improve the quality of our work.
>
> > ***W1**:Evaluation of realism by asking random people from social media if a scenario is realistic seems a bit weak in this particular case. It doesn't capture a lot of the nuances of these bullying dynamics. Also, it is not certain how valid their perception of what are or aren't realistic scenarios, especially if the participants did not even experience being bullied or have no experience with people who are bullied.*
>
> We acknowledge that relying on social media participants to assess the realism of bullying scenarios may not fully capture the nuanced dynamics of bullying situations. However, we adopted this approach after careful consideration, as a 2019 UNESCO global survey revealed that 32% of students worldwide experienced bullying in the month prior to the survey, with the highest bullying rates reaching 74% in some countries and regions[1]. This highlights the widespread nature of the bullying issue. The participants in our social media survey are likely to have experienced or witnessed varying degrees of bullying, and their insights into these scenarios provide valuable perspectives, which we believe are still reliable.
>
> Moreover, the definition of bullying itself is complex and subjective, often dependent on individual perceptions. Therefore, our approach aims to cover a broad range of experiences to account for this variability. We also recognize the ethical concerns about reaching out to students who have personally experienced bullying, as this could potentially retraumatize them. Given these considerations, we plan to seek expert evaluations from frontline teachers in future studies, who can provide a more informed and sensitive perspective on the realism of the simulations.
>
> [1]UNESCO. (2019). *Behind the numbers: Ending school violence and bullying*. Paris: UNESCO. Retrieved from https://unesdoc.unesco.org/ark:/48223/pf0000366486

---

> ### Author Response · Authors · 2025-11-21
> **Response to Reviewer yqFL [2/7]**
>
> > ***W2**:Evaluations also do not quite show convincing experiments or analysis on how well they capture the diversity of possible outcomes. They did mention that they ran 100 simulations with a wide range of behaviors and showed a table and a figure of a few representative outcomes in the appendix. However, there are no further analysis nor statistics shown on this distribution. It would be nice to see this and also validate how well the match distributions of reported cases.*
> >
>
> Thank you for raising this point. We would like to clarify that our submission already includes the requested distribution-level analysis in `Figure 9 of Appendix C` which summarizes the behavioral distributions aggregated over 100 independent simulations. Rather than showing only a few sample trajectories, `Figure 9` reports the **full cooperative vs. competitive action distribution** for each SVO profile (prosocial, individualistic, competitive, altruistic) across three different environments (study group, classroom collaboration, and leadership selection).
>
> These distributions explicitly capture the **diversity of possible outcomes** produced by the model. They demonstrate that agents with different SVO types generate systematically different behavioral frequency patterns, such as prosocial agents exhibiting higher cooperative rates and competitive agents producing more competitive actions, which is consistent with established SVO theory. This provides both **statistical evidence of outcome variability** and **theoretical validation** that the model’s generative distribution aligns with expected psychological constructs.
>
>
> >***W3**:  Also, it would be nice to see how the model handles/simulates rare behaviors where things might not go as expected. In some cases, it's precisely the rare behavior that we might want to simulate interventions.*
> >
>
> We agree that rare or unexpected behaviors are often the most important targets for educational intervention. In fact, **Case Study II is explicitly designed to evaluate the model’s capacity to simulate and respond to such low-frequency, high-impact events**. . Although most interactions remain cooperative, the simulation occasionally produces behaviors that drift away from expected norms, such as a student subtly exaggerating another candidate’s weaknesses during a conversation, or privately trying to sway a classmate’s vote at an inappropriate moment. The intervention strategies tested in Case II (e.g., cooperative reframing, fairness reminders) are specifically targeted at these unexpected competitive spikes rather than at routine interactions. Thus, the case serves as a direct demonstration of how EduMirror handles rare social behaviors and how the framework can be used to prototype and evaluate interventions when things do not go as expected.
>
> > ***W4**: There does not seem to be any evaluations done comparing with the actual reported bullying cases. I do acknowledge that this is difficult to collect, but without this data, it makes it difficult to assess realism.*
> >
>
> We also recognize that collecting real-world bullying data is challenging, as it often relies on self-reports and retrospective analysis, which can be limited by biases or incomplete information. To validate the realism of our simulation, we conducted a survey comparing simulated bullying scenarios with real-world incidents (see `Section 3.1`, “Evaluation of Simulation System” in the main text, and `Appendix H.2` for the full questionnaire). This comparison helped us verify the accuracy of our model in simulating common bullying behaviors.At the same time, we recognize that collecting data on sensitive bullying topics directly from students poses significant ethical challenges. This is precisely one of the reasons we focused on creating a bullying simulation model, offering a safer, more controlled way to study and intervene in bullying dynamics. As a next step, we plan to involve frontline educators in evaluating the realism of our simulations, which will provide us with more accurate insights and allow us to further refine the model.

---

> ### Author Response · Authors · 2025-11-21
> **Response to Reviewer yqFL [3/7]**
>
> > ***W5**:There are also no ablation studies or analysis of the design choices.*
> >
>
> Thank you for your comment regarding the absence of ablation studies and analysis of our design choices. We have now addressed this concern and incorporated a detailed ablation study into our work.
>
> （1）Specifically, for the bullying simulation, we use individual value-driven agents, with psychological needs modeled according to Maslow’s hierarchy of needs and the PERMA model from positive psychology. These models break down human psychological needs into five major categories and thirteen sub-dimensions, which we apply to our agents to drive diverse behaviors and reactions during interactions.
>
> Our ablation experiment investigates the effects of removing each category of psychological needs on the agent’s simulated behavior. The process involves running simulations with each category of psychological needs removed, while keeping the initial setup the same. We then compare these results with the full psychological needs-driven agent and have a large language model (LLM) rate the action sequences produced by the agents. The evaluations are based on three dimensions:
>
> - **Naturalness**: Refers to how well the behavior sequence aligns with the individual’s innate abilities, habits, and environmental context, reflecting authentic human psychological dynamics.
> - **Coherence**: Measures how logically and seamlessly different actions or steps in a sequence are integrated to achieve the intended goal, ensuring a consistent emotional progression.
> - **Plausibility**: Evaluates the rationality, possibility, or credibility of a sequence of actions, considering the environment, context, and known behavior patterns at the time.
>
> From this, we generated 50 sets of results and calculated the mean and standard deviation for each agent’s scores across the three evaluation dimensions. The results are as follows:
>
> | Agent | Lack of Safety Needs  |  | Lack of Self-Esteem Needs |  | Lack of Social Belonging Needs  |  | Lack of Meaning and Growth Needs |  | Lack of Psychological Health Needs  |  | Complete Psychological Needs  |  |
> | --- | --- | --- | --- | --- | --- | --- | --- | --- | --- | --- | --- | --- |
> |  | Mean | Std | Mean | Std | Mean | Std | Mean | Std | Mean | Std | Mean | Std |
> | Naturalness | 3.6 | 0.5292 | 3.12 | 0.8863 | 2.96 | 0.8237 | 3.84 | 0.8172 | 2.88 | 0.8635 | 4.56 | 0.5352 |
> | Coherence | 3.54 | 0.5370 | 3.1 | 0.9220 | 2.82 | 0.7922 | 3.72 | 0.7296 | 2.78 | 0.8553 | 4.34 | 0.5142 |
> | Plausibility | 3.56 | 0.5713 | 3.2 | 0.8485 | 2.92 | 0.7440 | 3.74 | 0.8762 | 2.86 | 0.7486 | 4.44 | 0.5713 |
>
> As shown in the table, the agent driven by the complete psychological need dimensions outperforms agents driven by the absence of any psychological need dimension in terms of naturalness, coherence, and plausibility. According to the scoring rationale provided by GPT-4o, agents driven by the complete psychological need dimensions are described as having multi-dimensional and layered emotional responses, demonstrating greater complexity and flexibility. This means that the agent is capable of producing more diverse and personalized reactions in a variety of situations, exhibiting stronger adaptability and emotional depth. In contrast, agents lacking any one of the psychological need dimensions tend to resort to avoidance strategies in complex situations, with emotional responses that are more mechanical and lacking in depth and variety. They display limited flexibility and diversity. Therefore, the complete psychological need-driven mechanism is better at simulating human emotional responses and behavioral reactions, offering more natural and human-like emotional expression.

---

> ### Author Response · Authors · 2025-11-21
> **Response to Reviewer yqFL [4/7]**
>
> > (Following) ***W5**:There are also no ablation studies or analysis of the design choices.*
> >
>
> （2）In Case 2, we examine how removing the Social Value Orientation (SVO)–driven mechanisms affects cooperative and competitive behaviors during social interactions. EduMirror agents are driven by both psychological needs and a dynamic SVO update mechanism, which enables them to adjust their social stance (e.g., altruistic, prosocial, individualistic, competitive) in response to interactions. This module encodes how agents trade off personal outcomes against others’ outcomes and is the key driver of social behaviors such as helping, coordinating, competing, or exploiting others.
>
> To evaluate the contribution of this mechanism, we conduct an ablation study by **removing SVO-related components**, while keeping all other parts of the system identical. The ablated agents thus rely only on internal desire fluctuations and general motivational tendencies, without personality-driven social preferences or SVO-based reasoning.
>
> We compare the full EduMirror model with the ablated version across four personality types. Large language models independently classify every action step into five categories: **Cooperation, Competition, Quasi-Cooperation, Quasi-Competition, Other.** The averaged results are shown below:
>
> | Personality | Model | Coop | Comp | Q-Coop | Q-Comp | Other |
> | --- | --- | --- | --- | --- | --- | --- |
> | Altruistic | Ours | 0.872 | 0.000 | 0.128 | 0.000 | 0.000 |
> |  | Ours w/o SVO | 0.891 | 0.000 | 0.109 | 0.000 | 0.000 |
> | Prosocial | Ours | 0.654 | 0.132 | 0.185 | 0.029 | 0.000 |
> |  | Ours w/o SVO | 0.875 | 0.074 | 0.035 | 0.016 | 0.000 |
> | Individualistic | Ours | 0.107 | 0.532 | 0.000 | 0.304 | 0.058 |
> |  | Ours w/o SVO | 0.126 | 0.636 | 0.007 | 0.204 | 0.027 |
> | Competitive | Ours | 0.040 | 0.783 | 0.000 | 0.177 | 0.000 |
> |  | Ours w/o SVO | 0.123 | 0.736 | 0.004 | 0.138 | 0.00 |
>
> In Case 2, removing the SVO mechanism leads to a clear contraction of behavioral patterns across all personality types. As shown in the table, altruistic and prosocial agents become uniformly cooperative, with substantially reduced quasi-cooperative and quasi-competitive behaviors, resulting in overly simplified and monotonic social responses. Conversely, individualistic and competitive agents collapse into narrowly focused competitive strategies, losing the mixed competitive and quasi-competitive patterns observed in the full model.
>
> These results indicate that SVO is essential for maintaining nuanced, personality-differentiated social behaviors. Without it, agents fall back into rigid, single-dimensional strategies, whereas the complete EduMirror model preserves richer intermediate behaviors and more realistic cooperation–competition dynamics.

---

> ### Author Response · Authors · 2025-11-21
> **Response to Reviewer yqFL [5/7]**
>
> > ***Q1**: Perception and belief update is not quite clear to me. Is it purely reliant on how the LLM wants to update this or is this somehow controlled or governed by some other formulation?*
> >
>
> In our framework, perception and belief update are not determined freely by the LLM; rather, they are governed by the structured reasoning pipeline explicitly defined in the paper. Perception is entirely driven by the environment, where each state $s_t$ produces an observation sequence $o_{0:t}$, forming the iterative loop: $st​⇒o0:t​⇒(Dti​,Dtj∣i​)⇒θti​⇒ati​⇒st+1​$, Thus, the LLM cannot choose “what to perceive”, it only interprets signals generated by the environment.Similarly, belief update is not freely generated but constrained by the decision formulation: $ati​∼Agent(⋅∣o0:t−1​,a0:t−1​,Dti​,θti​;ψi​)$.
>
> which ensures that internal belief evolution is tied to accumulated observations, motivational states $D_t$, and static agent characteristics $\psi_i$, not arbitrary LLM outputs. While the LLM contributes intermediate psychological assessments, these signals are embedded within deterministic update operators rather than becoming state updates themselves. For example: $D^{j|i}_t \sim LLM(o_{0:t-1}, a_{0:t-1})$, but the actual numerical motivational update is carried out by a fixed operator $U_D$, not by LLM text.
>
> To make this mechanism clearer, we have added detailed descriptions of the perception pipeline, belief-update operator, and motivational update process in `Section 2.2`.
>
> > ***Q2**：How is the memory managed? How do you decide what details to remember and what details to forget? Is it purely reliant on what the LLM chooses or do you store everything? Also, how does this compare with human recall?*
> >
>
> In our framework, the agent has both long-term memory and working memory. Long-term memory stores all experiences and perceptions, while working memory holds the states of multiple components, each reflecting the agent’s current tasks and context.
>
> **1. How is memory managed?**
>
> The agent’s memory is managed by several components. Each component has its own state and updates it based on new observations and the states of other components. Working memory holds these component states, while long-term memory stores more persistent information. When new information comes in, the relevant components in working memory update their states. For example, the goal component updates its state based on new observations, ensuring that memory updates align with the current task and context.
>
> **2. How do you decide what details to remember and what details to forget?**
>
> The decision of what to remember and forget is made by each component based on its current goal and context. Important and contextually relevant information is retained, while irrelevant or unnecessary information is forgotten. For instance, the goal component will prioritize retaining information that is closely related to achieving its objective, while less important details may be forgotten. This ensures that the memory is both efficient and relevant to the task at hand.
>
> **3. Is it purely reliant on what the LLM chooses, or do you store everything?**
>
> Memory updates are not solely dependent on the LLM’s choices. Each component decides, based on its current task and objective, which information to update or retain. The LLM’s role is to help summarize and reason over the incoming observations, but the decision-making about what gets stored or updated lies with the components. Components use their internal logic to decide whether to store or update information, rather than relying solely on the LLM.
>
> **4. How does this compare with human recall?**
>
> Compared to human memory, the agent’s memory is more structured and systematic. Human memory is often influenced by emotions, biases, and contextual factors, whereas the agent’s memory is managed by clear rules and interactions between components. As a result, the agent’s memory updates are more stable, predictable, and aligned with its goals. While human memory also has selective retention and forgetting, it is more prone to interference, whereas the agent’s memory is optimized according to predefined objectives and needs.

---

> ### Author Response · Authors · 2025-11-21
> **Response to Reviewer yqFL [6/7]**
>
> > ***Q3**: Are there analyses on what makes some of the generations unnatural both for the human and LLM evaluators? Do they align?*
> >
>
> We indeed conducted relevant analyses. Taking Case Study 1 simulated using the Individual Value Model as an example, after the survey experiment, we briefly interviewed five evaluators who had a high accuracy in selecting the real event (we did not inform them of the results). We asked them why they did not choose the other option (i.e., the simulated event). One respondent mentioned that the event lacked severe violent behavior, which made it different from bullying cases they had seen in the news and made it seem “unnatural.” Additionally, some of the simulated events showed the victim agent adjusting their mindset through practices like mindfulness, and the bully reflecting on their actions and stopping the bullying, which is less common in real-world experiences and contributed to the unnaturalness. Both the large model and human evaluators pointed out that the plot was more linear and lacked “rare behaviors” that might produce different outcomes, which also made the simulation feel unnatural.
>
> In analyzing the reasons for this, we identified two main factors:
>
> - First, our simulation is based on a LLM, and there are inherent value-alignment issues, which limit the occurrence of violent behavior and lead to “happy-ending” conclusions. However, we have attempted to control this issue by adding prompt words and trying different base LLMs, ultimately using DeepSeek-v3 as our base model.
> - Second, this study simplified the linear reasoning of human psychological needs changes, which cannot fully capture the complexity of real human psychological need evolution. Therefore, future research will expand the psychological needs-driven framework to simulate more complex events and actions, incorporating more layers of psychological needs as well as additional causal relationships and interactions into the modeling process.
>
> For Case Study 2, we further examined alignment between human and LLM evaluators using the same principle of comparing “naturalness judgments” across different personality-conditioned generations. Instead of focusing on specific events, we assessed whether humans and the LLM consistently preferred the same outputs as being more natural for a given personality setting. This comparison produced an agreement level of **0.77273**, indicating strong consistency between human evaluators and the LLM in identifying the more plausible behavioral trajectory. This suggests that the evaluation mechanism maintains good reliability across different types of scenarios.
>
> > ***Q4**：I can't seem to find validations on the dual track measurement protocol, how well does the llm rater and llm surveyor perform?*
> >
>
> Thank you for raising this critical point regarding the validation of our measurement tools. We apologize if these details were not immediately prominent in the main text. To address your concern, we provide the specific validation evidence for both tracks below.
>
> **1. Validation of the LLM Rater (Behavioral Assessment)**
> To validate the LLM Rater, we systematically compared its performance against human judgment to ensure reliability. As detailed in the **"Consistency Between Human Annotators and GPT-4o Evaluations"** of `Appendix E` , we conducted an experiment involving 15 human annotators who evaluated 20 randomly selected behavior sequences.
>
> - **Performance:** We compared the LLM's ratings with the human results. As shown in `Table 4` , for samples where human annotators reached high consensus (>75% agreement), the LLM Rater achieved **100% consistency** with human judgments. This empirical evidence confirms that the LLM Rater performs reliably and aligns closely with human perception in evaluating complex social behaviors.
>
> **2. Validation of the LLM Surveyor (Internal State Assessment)**
> For the LLM Surveyor, we ensure validity through **Construct Validity**. We strictly ground our probes in established, empirically validated psychometric scales rather than generating arbitrary questions.
>
> - **Method:** We operationalize abstract psychological constructs by adapting items from standard psychological inventories.
> - **Examples:**
>     - For **Self-Esteem**, we utilize the **Rosenberg Self-Esteem Scale (RSES)** [1] (e.g., probing the agent with: *"Do you feel that you have a number of good qualities?"*).
>     - For **Social Comparison**, we employ the **Iowa-Netherlands Comparison Orientation Measure (INCOM)** [2] (e.g., asking: *"How often do you compare what you have with what your friends have?"*).
>     This rigorous operationalization ensures that the internal states captured by the Surveyor are theoretically sound and consistent with established psychological research standards.

---

> ### Author Response · Authors · 2025-11-21
> **Response to Reviewer yqFL [7/7]**
>
> > ***Q5**: I also cant seem to find experiments / analysis on the different settings / virtual environments.*
> >
>
> Thank you for raising this point. As illustrated in the `Figure 1`, our simulator supports a wide range of typical school environments, including classrooms, hallways, playgrounds, and dormitories. We ran simulations across these settings to better capture the complexity of real school social contexts. Due to space limitations, detailed environment-specific cases are placed in the appendix. For example, `Table 7 in Appendix G.2` presents bullying behavior simulated across different settings.
>
> In addition, **Case Study II** explicitly varies the **social scale** of interaction through three educational scenarios drawn from our scenario library:
>
> 1. A dyadic study partner setting with 2 agents;
> 2. A small study group with 4 agents engaging in close peer interaction and resource sharing;
> 3. A class-wide leadership election with 20 agents involving public speeches, alliance formation, and direct vote competition.
>
> These three settings form a spectrum of increasing social complexity, and we systematically log cooperative and competitive actions under each configuration. The aggregated distributions reported in `Figure 9` in the `Appendix B` show that, across these different social scales, EduMirror consistently produces SVO-consistent behavior patterns while also capturing shifts in cooperation–competition frequencies as group size and interaction structure change.
>
> [1]Rosenberg, M. Society and the Adolescent Self-Image. Princeton University Press, 1965.
>
> [2]Gibbons, F. X., & Buunk, B. P. Individual differences in social comparison: Development of a scale of social comparison orientation. Journal of Personality and Social Psychology, 1999.

---

### Official Review · Reviewer_dp4X · 2025-10-30

**Soundness:** 3
**Presentation:** 3
**Contribution:** 3
**Rating:** 6
**Confidence:** 3

**Summary:**

This paper introduces EduMirror, a modular multi-agent simulation platform designed to model and experimentally analyze educational social dynamics such as bullying, peer pressure, and group cooperation.The system comprises four components: a systematic scenario design workflow grounded in social science theory, a value-driven agent architecture encompassing both individual psychological needs and social value orientation, a dual-track measurement protocol that transforms qualitative behavioral interactions into quantitative data via LLM-based assessors, and tools for user-driven intervention and branching causal experimentation. The authors conduct two case studies—simulating school bullying and peer group cooperation—demonstrating its capacity to generate theoreticallyconsistent and empirically-verifiable social phenomena.

**Strengths:**

1.The scene design workflow and agent architecture are explicitly grounded in established social science theories and can be flexibly tailored for diverse scenarios. Furthermore, to address challenging-to-measure behavioral and psychological dynamics, this paper proposes a dual-track measurement approach that leverages two LLMs working in tandem to convert these qualitative interactions into quantifiable data.

2.This paper presents an innovative integration of psychological theory, social science theory, and LLM measurement tailored for educational settings, and enabling interpretable, relevant experimentation. Experiments demonstrate that edumirror outperforms the baseline comparison.

**Weaknesses:**

1.This paper demonstrates the effectiveness of EduMirror through two case studies. While the case designs align with the research theme, they do not fully prove that EduMirror can robustly generalize across different educational systems, age groups, or larger social networks. This limitation somewhat restricts the method's broader applicability.

2.This architecture implicitly relies on the psychological validity of value-driven models and  LLM interpretability and ability, but does not provide formal guarantees. It is recommended to add theoretical foundations to support the paper's arguments.

3.The current experimental setup fails to fully demonstrate EduMirror's strengths. The choice of benchmarks for comparison (such as reAct and BabyAGI) appears somewhat outdated.

**Questions:**

The paper provides limited technical details regarding the construction of EduMirror. Could this section be supplemented with additional information (such as the LLM prompts and scoring rubric)?

---

> ### Author Response · Authors · 2025-11-21
> **Response to Reviewer dp4X[1/2]**
>
> We sincerely thank the reviewer for the thoughtful evaluation and constructive suggestions. We appreciate the recognition of our framework, theoretical grounding, and experimental design, and we address each comment in detail below.
>
> > ***W1**: This paper demonstrates the effectiveness of EduMirror through two case studies. While the case designs align with the research theme, they do not fully prove that EduMirror can robustly generalize across different educational systems, age groups, or larger social networks. This limitation somewhat restricts the method's broader applicability.*
>
> Our original submission focused on two representative phenomena, bullying and peer cooperation, as proof-of-concept demonstrations. Following the reviewer’s suggestion, we have significantly expanded our evidence for generalizability through **two additions**:
>
> - **University**: This scenario shows students navigating lectures, study spaces, and peer interactions while managing academic pressure and personal goals.
> - **Family**: This scenario depicts children completing homework under parental supervision, balancing expectations, routines, and emotions.
>
> In the university scenario, agents demonstrated coherent academic behaviors as they coordinated study tasks, negotiated roles within the group, and responded appropriately to collaboration successes and minor coordination challenges. In the family scenario, the interactions reflected common parent–child learning patterns, with children alternating between working on homework, seeking approval, and managing emotions while the parent provided guidance and structure. Across these additional settings, the system generated socially plausible interaction patterns that are consistent with those observed in our original classroom studies, supporting the broader applicability of EduMirror’s value-driven architecture.
>
> We evaluate Naturalness and Human-likeness on a **5-point scale**, and the results of the new simulations are summarized below:
>
> | Method | University |  | Family |  | Classroom | |
> | --- | --- | --- | --- | --- | --- | --- |
> |  | Naturalness | Human-likeness | Naturalness | Human-likeness  | Naturalness | Human-likeness |
> | ReAct | 3.333 | 3.625 | 3.700 | 3.933 | 3.950 | 4.208 |
> | BabyAGI | 3.792 | 3.875 | 3.800 | 4.008 | 3.958 | 3.958 |
> | LLMob | 3.958 | 4.000 | 3.702 | 4.000 | 4.042 | 4.333 |
> | D2A | 3.803 | 4.417 | 3.792 | 3.958 | 3.867 | 4.117 |
> | JAG-concordia | 4.042 | 4.250 | 4.000 | 4.167 | 4.083 | 4.192 |
> | **Ours** | **4.625** | **4.667** | **4.517** | **4.642** | **4.708** | **4.824** |
>
> Across the two additional scenarios, the results for Naturalness and Human-likeness closely mirror those reported in our original two case studies. This consistency indicates that EduMirror’s value-driven agent architecture generalizes reliably across different settings and continues to generate psychologically plausible, human-aligned behavior beyond the initial examples.

---

> ### Author Response · Authors · 2025-11-21
> **Response to Reviewer dp4X[2/2]**
>
> > ***W2**: This architecture implicitly relies on the psychological validity of value-driven models and LLM interpretability and ability, but does not provide formal guarantees. It is recommended to add theoretical foundations to support the paper's arguments.*
>
> Thank you for the comment. While the framework does not offer formal guarantees, its design is grounded in well-established theories of human motivation rather than unconstrained LLM behavior. Need-based motivational theories, from Maslow’s hierarchy to more recent formulations such as Self-Determination Theory, model human action as driven by the fulfillment of core physiological and psychological needs[1,2]. The value dimensions in EduMirror are a task-specific operationalization of these latent needs: they distinguish foundational psychological motives and are further refined with domain experts depending on the educational scenario, ensuring that the value vector is theoretically meaningful rather than arbitrarily chosen.
>
> Moreover, the idea of representing internal needs as latent value variables and using them to drive agent behavior has been empirically validated in prior work on desire- and value-driven LLM agents, such as the D2A framework[3], where need-based internal states were shown to produce more coherent and human-like daily activity patterns than prompt-only baselines. EduMirror extends this validated modeling strategy from everyday activities to educational social dynamics and couples it with a structured measurement protocol, so the architecture stands on an existing line of need-based motivational theory and empirically tested value-driven agents, rather than introducing an entirely new or unsupported psychological model.
>
> We also emphasize that EduMirror is not limited to a single case: as shown in the newly added  `Table 2 of Appendix C` , the framework supports a rich library of 20 theory-grounded educational scenarios and can be extended to additional settings, demonstrating that our value-based modeling generalizes well beyond the reported case studies.
>
> References:
>
> [1] Maslow, A. H. *A Theory of Human Motivation*. Psychological Review, 1943.
>
> [2] Ryan, R. M., & Deci, E. L. *Self-Determination Theory and the Facilitation of Intrinsic Motivation, Social Development, and Well-Being*. American Psychologist, 2000.
>
> [3]Wang, Y., Chen, Y., Zhong, F., Ma, L., & Wang, Y. (2024). Simulating human-like daily activities with desire-driven autonomy. *arXiv preprint arXiv:2412.06435*. https://arxiv.org/abs/2412.06435
>
> > ***W3**: The current experimental setup fails to fully demonstrate EduMirror's strengths. The choice of benchmarks for comparison (such as reAct and BabyAGI) appears somewhat outdated.*
>
> To better highlight EduMirror’s capabilities,
> we conducted additional comparisons using two stronger and more relevant baselines: **D2A**, a widely used LLM-agent framework, and **JAG**, the first-place agent system in the Concordia competition.
> We evaluated them in both case studies to provide a clearer picture of our strengths. The results are summarized in the tables below.
>
> Case Study 1 Comparison
>
> |   | Ours |  |   D2A  |  |  JAG |  |
> | --- | --- | --- | --- | --- | --- | --- |
> |  |  Mean | Std | Mean | Std  |  Mean | Std |
> | Naturalness | **4.24** | **0.5911** | 3.76 | 0.7969 | 3.52 | 0.7068 |
> | Coherence | **4.56** | **0.5014** | 4.06 | 0.6518 | 4.16 | 0.8657 |
> | Plausibility | **4.48** | **0.5436** | 3.86 | 0.9478 | 3.78 | 0.8401 |
>
> Case Study 2 Comparison
>
> | Model | **DeepSeek** |  | **GPT-4.1** |  | **Gemini** |  | **Qwen3** |  |
> | --- | --- | --- | --- | --- | --- | --- | --- | --- |
> |  |  Naturalness | Human-likeness |  Naturalness | Human-likeness |  Naturalness | Human-likeness |  Naturalness | Human-likeness |
> | D2A | 3.925 | 4.100 | 3.700 | 3.933 | 4.042 | 4.181 | 3.867 | 4.117 |
> | JAG | 3.760 | 3.875 | 3.958 | 4.133 | 3.885 | 4.052 | 4.083 | 4.192 |
> | **Ours** | **4.750** | **4.792** | **4.958** | **4.958** | **4.708** | **4.708** | **4.792** | **4.824** |
>
> These results show that EduMirror consistently performs at a higher level across dimensions aligned with our research goals.
>
> >***Q**: The paper provides limited technical details regarding the construction of EduMirror. Could this section be supplemented with additional information (such as the LLM prompts and scoring rubric)?*
>
> Thank you for the helpful suggestion. We have incorporated the requested technical details into the supplementary materials and added the relevant LLM prompts and scoring rubrics to `Appendix H` for clarity and reproducibility.

---

### Official Review · Reviewer_Erqf · 2025-11-10

**Soundness:** 3
**Presentation:** 3
**Contribution:** 3
**Rating:** 6
**Confidence:** 3

**Summary:**

This paper proposes a simulation platform to model educational social dynamics with workflow grounding, value-driven agent model, and LLMs based rater for valuation. Cases studies on bullying and group cooperation are conducted to validate the platform.

**Strengths:**

• A systematic modularized architecture for social dynamics simulation
	• A LLM-based measurement protocol to harvest the benefits of latest advancement in LLMs to serve social dynamic evaluation in simulation
	• Human evaluation are conducted to verify whether the simulated cases are indistinguishable from real cases from the human points of view
	• Individual value models are evaluated using LLM models and comparing to human judges' evaluation to confirm alignment
Agent models are theoretical grounded: based on the theories of psychological needs and social value models.

**Weaknesses:**

• The intervention stratigies lack diversities with only 3 strategies
	• The agent models (with their interaction with external world) are based on designated theories in a classic manner: How agent customization and how the intervention strategies affects the actitivies and event sequences progression are governed by classific rule-based theories only lacking diversities as real human behaviors. The problem of how to reconcile the benefits of LLMs to generate diversified and plausbile behaviors while being realistic (either according to ceterin well-verified theories or realsitic dataset) is not solved.

**Questions:**

1. Latex issue on left quotation marks: e.g. line 052 and a few other places;   `` should be used instead  of ''
	2. Line 200-207, please explain or re-iterate the argument in the literature regardin ghow value-driven agent achitecture grounded in pyschological theory and dual-track measurement protocol can help to releive the LLMs' generative approach from rigorous validity issue. There seems missing a logical link to connect the value-driven models and rigorous experiments. Also please explicitly discuss what have been traded (e.g. whether the diversity or human-like behavior) if any by removing LLM-based simluation for value-driven theory validity?
	3. Line 320, what exactly are the choices human participants have? Only two choices: "real" and "diffifculity to distinguish"? Any bias towards designated answers in the question design?
	4. Line 380-383, please provide more details on how teacher agents and the student agents interact w.r.t. different intervention and context. Also how to verify the validity, fidelity and branch coverage (diversity coverage) of the intervention and the intervention impact that are generated?

---

> ### Author Response · Authors · 2025-11-21
> **Response to Reviewer Erqf[1/2]**
>
> Thank you for your constructive review. We appreciate your recognition of our modular architecture, the LLM-based measurement protocol, and the theoretical grounding of our agent models. We have addressed your specific concerns regarding intervention diversity, the balance between theory and LLM capabilities, and experimental details below.
>
> > ***W1**: The intervention strategies lack diversity with only 3 strategies.*
>
> Regarding the concern that “the intervention strategies lack diversity and only include three strategies,” we would like to clarify that in Case Study 1, the strategies are not limited to just three, but are categorized based on established classical literature [1]. During the experiment, teacher agents with these three different intervention strategies, as well as a control teacher agent (which completely ignores bullying), autonomously generated a wide variety of specific intervention behaviors, as shown in `Table 10 of Appendix G.5`. These behaviors reflect the practical application of the different strategies and provide valuable insights for real-world teacher interventions. Similarly, in Case Study 2, our experiment is also not restricted to three interventions. The three presented in the paper were selected because they are representative strategies commonly discussed in educational research, enabling us to verify that EduMirror can model how different interventions influence the emergence or mitigation of malicious competition.
>
> Additionally, it is important to note that the purpose of the intervention experiments is to demonstrate the platform’s ability to simulate and analyze the effects of different intervention strategies in educational contexts, rather than to explore every possible strategy. These interventions are intended as illustrative examples, not limitations of the system. EduMirror can easily accommodate additional or customized intervention types.
>
> [1] Seidel A, Oertel L. A categorization of intervention forms and goals[M/OL]//Bilz L, Schubarth W, Dudziak I, et al. Gewalt und Mobbing an Schulen. Wie sich Gewalt und Mobbing entwickelt haben, wie Lehrer intervenieren und welche Kompetenzen sie brauchen. Bad Heilbrunn,Germany: Klinkardt, 2017: 13-25. DOI: 10.1086/428763.
>
> > ***W2:** The agent models (with their interaction with external world) are based on designated theories in a classic manner: How agent customization and how the intervention strategies affects the actitivies and event sequences progression are governed by classific rule-based theories only lacking diversities as real human behaviors. The problem of how to reconcile the benefits of LLMs to generate diversified and plausbile behaviors while being realistic (either according to ceterin well-verified theories or realsitic dataset) is not solved.
> **Q2:** Line 200-207, please explain or re-iterate the argument in the literature regardin ghow value-driven agent achitecture grounded in pyschological theory and dual-track measurement protocol can help to releive the LLMs' generative approach from rigorous validity issue. There seems missing a logical link to connect the value-driven models and rigorous experiments. Also please explicitly discuss what have been traded (e.g. whether the diversity or human-like behavior) if any by removing LLM-based simluation for value-driven theory validity?*
> >
>
> We respectfully clarify that EduMirror is not a rule-based system governed by rigid scripts, but a Value-Driven Generative Architecture.
>
> 1. **Theory as a Guide, Not a Script:** Psychological theories (e.g., Needs, SVO) define the agent's *internal state* (motivation), but the LLM functions as the reasoning engine to generate open-ended, context-aware actions. As shown in `Figure 10 (Appendix D)`, the "Value-Driven Planner" uses theory to evaluate options, but the generation relies on the LLM.
> 2. **Logical Link to Validity:** The "validity issue" in standard LLM simulations stems from their "black box" nature. EduMirror resolves this by:
>     - **Internal Validity:** Explicitly modeling motivation (e.g., tracking a "Safety" score) ensures behaviors are causally linked to psychological states, not random hallucinations.
>     - **Construct Validity:** The "LLM Surveyor" (Dual-Track Protocol) transforms behaviors into standardized psychometric data , aligning simulation data with empirical social science.
> 3. **The Trade-off:** We have traded **"unconstrained stochasticity"** (noise) for **"psychological consistency"** (validity). Without theoretical grounding, pure LLM agents often exhibit "schizophrenic" behavior, acting inconsistently with their persona. By constraining the LLM with values, we lose some randomness but gain **character depth**. As shown in **Figure 5**, our method consistently outperforms pure LLM baselines (ReAct, BabyAGI) in human-likeness and naturalness, proving that this trade-off enhances rather than limits realism.

---

> ### Author Response · Authors · 2025-11-21
> **Response to Reviewer Erqf[2/2]**
>
> > ***Q3:** Line 320, what exactly are the choices human participants have? Only two choices: "real" and "diffifculity to distinguish"? Any bias towards designated answers in the question design?*
>
>
> Thank you for your question. We have included the complete questionnaire in `Appendix H.2` for your convenience. In the questionnaire design for human participants, there are indeed three options provided for them to choose from: “1 is real”, “2 is real”, and “difficult to distinguish.” After each paired case comparison, the participants are asked to select the one they believe is closer to reality, or choose “difficult to distinguish.”
>
> Regarding the survey setup process, this study first collected 10 real bullying cases from the internet, news reports, and interviews with friends. Then, we selected 10 simulated cases from a pool of 100 generated bullying scenarios that were similar to these 10 real cases in terms of scene setting and storyline content. To avoid participants distinguishing real and simulated cases based on language style, we used a large language model to extract key events from each case and rephrase them in a unified style. This ensured that both types of cases were consistent in terms of word count, language style, and narrative structure, thus eliminating any potential bias. Therefore, we believe the design of our questionnaire does not lead to biased choices from participants.
>
> > ***Q4:** Line 380-383, please provide more details on how teacher agents and the student agents interact w.r.t. different intervention and context. Also how to verify the validity, fidelity and branch coverage (diversity coverage) of the intervention and the intervention impact that are generated?*
>
> In the school bullying intervention experiment, the bully agent and the victim agent interact autonomously to simulate bullying incidents. During this process, the teacher’s goal is to intervene using one of the three strategies upon observing a bullying event. Additionally, this study sets up a control group to simulate the case where the teacher completely ignores the bullying incident. During the experiment, teacher agents with different intervention strategies autonomously generate a variety of intervention behaviors based on the current context, as shown in `Table 10 of Appendix G.5`.
>
> For example, under the “Supportive-Individual” intervention strategy, the teacher may engage in a one-on-one conversation with the victim. At each simulation time step, the actions of all parties are summarized into an “observation” and returned to each agent. The victim agent, driven by the Individual Value Model, generates responses and behaviors that best align with its current psychological needs, based on contextual information and its psychological need values, and updates its psychological need values accordingly. For instance, the victim agent may cry and share their bullying experience during a one-on-one conversation with the teacher. Various factors, such as the initial scenario, the teacher’s intervention strategy, and the agent’s psychological need values, influence the progression of the intervention simulation. These factors contribute to the complex dynamics of simulating real-life school bullying interventions.
>
> We verified the simulation's validity by comparing the outcomes with established educational psychology literature. Our results (`Figure 6`) show that the *Supportive-Cooperative* strategy is the most effective for victim recovery, while *Authoritative* interventions have limited effects. This alignment with empirical findings [1] confirms the ecological validity of the generated interactions. Regarding branch coverage, it is important to note that we do not intend to explore all possible intervention strategies. As mentioned earlier, the purpose of the intervention experiment is to showcase the platform’s ability to simulate and analyze the effects of different strategies in educational contexts. We hope that future researchers will be able to use this platform to explore the effectiveness of various intervention strategies.
>
> [1] Wachs, Sebastian, et al. "Bullying intervention in schools: A multilevel analysis of teachers’ success in handling bullying from the students’ perspective." *The Journal of Early Adolescence* 39.5 (2019): 642-668.
>
> > ***Q1:** Latex issue on left quotation marks: e.g. line 052 and a few other places; `` should be used instead of ''*
>
> Thank you for spotting this. We have corrected the quotation mark errors (using `` for opening quotes) and reviewed the manuscript to ensure all formatting complies with standard LaTeX practices.

---

### Author Response · Authors · 2025-11-21
**Revision Note**

We would like to **thank the reviewers for their thoughtful comments and constructive feedback**. We have carefully considered all the points raised and made several revisions to address the concerns. Below, we summarize the **key changes** made in response to the reviewers’ suggestions:

1. The **open-source code** for the EduMirror platform has been released, and the link is as follows: **https://anonymous.4open.science/r/EduMirror**
2. We have added **ablation experiments**. In Case Study 1, we tested the impact of different dimensions of psychological needs on the performance of the Individual Value model, with detailed information provided in **`F.3`**. In Case Study 2, we tested the effect of different SVO groups on the performance of the Social Value model, with details provided in **`Appendix D.3`**.
3. Two **updated and stronger baseline models, D2A[1] and JAG-Concordia[2]**, have been introduced, and benchmark tests have been conducted for both the Individual Value model and the Social Value model. Specific details and results have been added to **`Section 3.2`** and **`Appendix G.3`** of the paper.
4. More **technical details** have been added to **`Section 2`, “EDUMIRROR,”** to help readers better understand the system, including the core architecture of the agent, system update rules, and decision-making processes.
5. **`Appendix C`** has been added, explaining the **pre-designed scenarios in EduMirror**, including examples, scenario expansion and generalization, and a complete description of the scenario library.
6. **`Appendix H`** has been added, containing the natural language **prompts for the core driving process of the agent** and the **full questionnaire** from the bullying simulation experiment in Case Study 1.

References:

[1]Wang, Y., Chen, Y., Zhong, F., Ma, L., & Wang, Y. (2024). Simulating Human-like Daily Activities with Desire-driven Autonomy. *ArXiv, abs/2412.06435*.

[2]Jordine, J. (2024). *jag-concordia: Agent files for the Concordia contest* [GitHub repository]. Winner of the JAG-Concordia entry for the Concordia Contest 2024. https://github.com/Jordine/jag-concordia

---

### Note · Program_Chairs · 2026-01-17
**Submission Desk Rejected by Program Chairs**

The following references in this submission do not refer to real documents and/or have major errors in bibliographic information:

 A. Seidel and L. Oertel. A categorization of intervention forms and goals. In L. Bilz, W. Schubarth, I. Dudziak, et al. (eds.), Gewalt und Mobbing an Schulen. Wie sich Gewalt und Mobbing entwickelt haben, wie Lehrer intervenieren und welche Kompetenzen sie brauchen, pp. 13-25. Klinkhardt, Bad Heilbrunn, Germany, 2017. doi: 10.1086/428763. [M/OL]

Zeyu Wang, Xing Xie, Junzhe Zhu, et al. Llmob: Large language model as socially embodied agents. arXiv preprint arXiv:2402.00168, 2024c.